



## Regional CO₂ Fluxes during 2010-2015 Inferred from GOSAT XCO₂

## retrievals using a new version of Global Carbon Assimilation System

Fei Jiang[1,7*], Hengmao Wang[1], Jing M. Chen[2], Weimin Ju[1], Xiangjun Tian[3], Shuzhang Feng[1], Guicai Li[4], Zhuoqi Chen[5], Shupeng Zhang[5], Xuehe Lu[1], Jane Liu[2,6], Haikun Wang[6], Jun Wang[1], Wei He[1], Mousong Wu[1]

*1 Jiangsu Provincial Key Laboratory of Geographic Information Science and Technology, International Institute for Earth System Science, Nanjing University, Nanjing, 210023, China*

*2 Department of Geography and Planning, University of Toronto, Toronto, Ontario M5S3G3, Canada*

*3 The Institute of Atmospheric Physics, Chinese Academy of Sciences, Beijing, 100029, China*

*4 National Satellite Meteorological Center, China Meteorological Administration, Beijing 100101, China*

*5 College of Global Change and Earth System Science, Beijing Normal University, Beijing, 100875, China*

*6 School of Atmospheric Sciences, Nanjing University, Nanjing, 210023, China*

*7 Jiangsu Center for Collaborative Innovation in Geographical Information Resource Development and Application, Nanjing, 210023, China*

* Corresponding author: Tel.: +86-25-83597077; Fax: +86-25-83592288; E-mail address: jiangf@nju.edu.cn





**Abstract**
Satellite $XCO_2$ retrievals could help to improve carbon flux estimation because of
their good spatial coverage. In this study, to assimilate the GOSAT $XCO_2$ retrievals,
the Global Carbon Assimilation System (GCAS) is upgraded with new assimilation
algorithms, procedures and a localization scheme, a higher assimilation parameter
resolution and so on, and hence is named as GCASv2. Based on this new system, the
global terrestrial ecosystem (BIO) and ocean (OCN) carbon fluxes from May 1, 2009
to Dec 31, 2015 are constrained using the GOSAT ACOS $XCO_2$ retrievals (Version 7.3).
The posterior carbon fluxes from 2010 to 2015 are independently evaluated using $CO_2$
observations from 52 surface flask sites. The results show that the posterior carbon
fluxes could significantly improve the modeling of atmospheric $CO_2$ concentrations,
with global mean BIAS decreases from a prior value of 1.6±1.8 ppm to -0.5±1.8 ppm.
Globally, the mean annual BIO and OCN carbon sinks and their interannual variations
inferred in this study are very close to the estimates of CT2017 during the study period,
and the inferred mean atmospheric $CO_2$ growth rate and its interannual changes are also
very close to the observations. Regionally, over the northern lands, there are the
strongest carbon sinks in North America Temperate, followed by Europe, Boreal Asia,
and Temperate Asia; and in tropics, there are strong sinks in Tropical South America
and Tropical Asia, but a very weak sink in Africa. This pattern is significantly different
from the estimates of CT2017, but the estimated carbon sinks in each continent and
some key regions like Boreal Asia and Amazon are comparable or in the range of
previous bottom-up estimates. The inversion also changes the interannual variations of
carbon fluxes in most TRANSCOM land regions, which have a better relationship with
the changes of severe drought area or LAI, or are more consistent with previous
estimates for the impact of drought. These results suggest that the GCASv2 system
works well with the GOSAT $XCO_2$ retrievals, and has a good performance in estimating
the surface carbon fluxes, meanwhile, our results also indicate that the GOSAT $XCO_2$
retrievals could help to better understand the interannual variations of regional carbon
fluxes.



## 1. Introduction

Atmospheric carbon dioxide ($CO_2$) is one of the most important greenhouse gases, and fossil fuel burning and land use change are mostly responsible for its increase from the preindustrial concentration. Terrestrial ecosystems and oceans play very important roles in regulating the atmospheric $CO_2$ concentration. In the past half century, about 60% of the anthropogenic $CO_2$ emissions have been absorbed by the terrestrial ecosystems and oceans (IPCC, 2014). However, their carbon uptakes have significant spatial differences and inter-annual variations. Therefore, it is very important to quantify the global and regional carbon fluxes.

Atmospheric inversion is an effective method for estimating the surface $CO_2$ fluxes using the globally distributed atmospheric $CO_2$ concentration observations (Enting and Newsam, 1990; Gurney et al., 2002). It has robust performance on global or hemisphere scale carbon budget estimates (Houweling et al., 2015), but on regional scales, due to the uneven distribution of in situ observations, the reliability of inversion results varies greatly in different regions. Generally, the inversions have large uncertainties in tropics, southern hemisphere oceans and most continental interiors such as South America, Africa, and Boreal Asia (Peylin el al., 2013). Satellite observation has a better spatial coverage, especially over remote regions, and studies show that it can be used to improve the carbon flux estimates (e.g., Chevallier et al., 2007; Miller et al., 2007; Hungershoefer et al., 2010). The Greenhouse Gases Observing Satellite (GOSAT) (Kuze et al., 2009), being the first satellite mission dedicated to observing $CO_2$ from space, was launched in 2009. Many inversions have utilized the GOSAT retrievals for column-averaged dry air mole fractions of $CO_2$ ($XCO_2$) to infer surface carbon fluxes (e.g., Basu et al., 2013; Maksyutov et al., 2013; Saeki et al., 2013a; Chevallier et al., 2014; Deng et al., 2014; Deng et al, 2016; Wang et al., 2018; Wang et al., 2019). Takagi et al. (2011) found that GOSAT $XCO_2$ retrievals could significantly reduce the uncertainties in estimates of surface $CO_2$ fluxes for regions in Africa, South America, and Asia, where the sparsity of the surface monitoring sites is most evident. Basu et al. (2013) shown that assimilating only GOSAT data can well reproduce the observed $CO_2$



time series at the surface and TCCON sites in the tropics and the northern extra-tropics,
but enhance seasonal cycle amplitudes in the southern extra-tropics. Wang et al. (2019)
also showed that GOSAT $XCO_2$ retrievals can effectively improve carbon flux
estimation, and the performance of the inversion with GOSAT data only was
comparable with the one using in situ observations. Meanwhile, based on the inversions
with GOSAT $XCO_2$ retrievals, Liu et al. (2018) quantified the impacts of the 2011 and
2012 droughts on terrestrial ecosystem carbon uptake anomalies over the contiguous
US, Deng et al. (2016) compared the distributions of drought and posterior carbon
fluxes in South America for 2010-2012, Detmers et al. (2015) studied the impact of the
strong La Niña episode on the carbon fluxes in Australia from the end of 2010 to early
2012. However, so far, on the one hand, most studies focused on the impact of GOAST
$XCO_2$ retrievals on the inversion of surface carbon fluxes, but in many regions, there
are still large divergences for carbon sinks between different inversions with the same
GOSAT data or between inversions with GOSAT and in situ observations (Chevallier
et al., 2014), on the other hand, although some studies reported the impact of drought
or extreme wetness on the changes of carbon fluxes using inversions based on GOSAT,
few studies have comprehensively investigated the impacts of GOSAT data on the
interannual variations of inverted land sinks in different regions.

In this study, we present a 6-year inversion from 2010 to 2015 for the global and

regional carbon fluxes using only the GOSAT $XCO_2$ retrievals. The Global Carbon
Assimilation System (GCAS) is employed to conduct this inversion, which was
developed in China in 2015 (Zhang et al., 2015) and updated in this study with a new
scheme to assimilate $XCO_2$ retrievals. The inverted multi-year averaged carbon fluxes
for the globe, global land and ocean, each TRANSCOM region as well as some key
areas are shown and compared with previous top-down and bottom-up (Jiang et al.,
2016) estimates. The estimated interannual variations of carbon fluxes in each
TRANSCOM region are given and discussed against changes in drought and LAI. This
manuscript is organized as follows: Section 2 details the GCASv2 system as well as the
prior fluxes, GOSAT retrievals and uncertainty settings. Section 3 briefly introduces the



experimental design. Results and discussions are presented in Section 4, and
Conclusions are given in Section 5.

## 2. Method and Data

### 2.1 A new version of the Global Carbon Assimilation System (GCASv2)

Figure 1 shows the flow chart of the GCASv2 system. In each data assimilation
(DA) window, there are two steps. The first step, the prior fluxes of $X^b$ are perturbed
with a Gaussian random distribution, and put into the global atmospheric chemical
transport model MOZART-4 to simulate $CO_2$ concentrations, which are then sampled
according to the locations and times of $CO_2$ observations. The sampled data are used in
the assimilation module together with the $CO_2$ observations to generate the optimized
fluxes of $X^a$. In the second step, the MOZART-4 model is run again using the
optimized fluxes of $X^a$, to generate new $CO_2$ concentrations for the initial field of the
next DA window. This DA flow chart is different from the previous version of GCAS,
which runs the MOZART-4 model only once, and optimizes the fluxes and the initial
field of the next window synchronously. In this study, we find the synchronous dual
optimizations will weaken the assimilation benefits on fluxes.
The perturbation of $X^b$ represents the uncertainty of the prior carbon flux, which
is calculated using the following function.
$$X_i^b = X_0^b + \lambda \times \delta_i \times X_0^b \quad , i = 1, 2, \dots, N \qquad (1)$$
where $\delta_i$ represents random perturbation samples, which is drawn from Gaussian
distributions with mean zero and standard deviation of one. N is the ensemble size. λ is
a set of scaling factors, which represents the uncertainty of each prior flux. In previous
version GCAS, λ is defined in different land and ocean areas based on 22 TRANSCOM
regions (Gurney et al., 2002) and 19 Olson ecosystem types, as in CarbonTracker
(Peters et al., 2007). This means that in the same area, the error of a prior flux is the
same. Through assimilation, the flux will be integrally enlarged or reduced. In GCASv2,
we change to use a λ in each grid, meaning that for each grid, the perturbations of prior
fluxes are independent. In addition, the grid cell of λ is different from those of the prior
flux and the transport model, which could be set freely.



Generally, there are 4 types of carbon fluxes, namely terrestrial ecosystem (BIO)
carbon flux, atmosphere and ocean (OCN) carbon exchange, fossil fuel (FOSSIL)
carbon emission and biomass burning (FIRE) carbon emission, which are used to drive
the transport model to simulate the atmospheric $CO_2$ concentration. And in general,
FOSSIL and FIRE fluxes are assumed to have no errors, only BIO and OCN fluxes are
optimized in an assimilation system (e.g., Peters, et al., 2007; Jiang et al., 2013; Wang
et al., 2019). In GCASv1, only the BIO flux was optimized, the OCN flux was directly
from the output of CarbonTraker (CT). In GCASv2, it is set to be an optional item. Four
schemes are set (Functions 2 - 5). The first one is the same as the previous version, only
the BIO flux is optimized; the second one is the same as general, namely both BIO and
OCN fluxes are optimized; the third one is that BIO, OCN and FOSSIL fluxes are
optimized at the same time; and the fourth one is that only net flux is optimized.
$$X_i^b = (X_{bio}^b + \lambda_{bio} \times \delta_{i,bio} \times X_{bio}^b) + X_{ocn}^b + X_{fossil}^b + X_{fire}^b, \text{ i} = 1, 2, \dots, N \quad (2)$$

$$X_i^b = \left(X_{bio}^b + \lambda_{bio} \times \delta_{i,bio} \times X_{bio}^b\right) + \left(X_{ocn}^b + \lambda_{ocn} \times \delta_{i,ocn} \times X_{ocn}^b\right)$$

$$+ X_{fossil}^b + X_{fire}^b, \text{ i} = 1, 2, \dots, N \quad (3)$$

$$X_i^b = \left(X_{bio}^b + \lambda_{bio} \times \delta_{i,bio} \times X_{bio}^b\right) + \left(X_{ocn}^b + \lambda_{ocn} \times \delta_{i,ocn} \times X_{ocn}^b\right)$$

$$+ (X_{fossil}^b + \lambda_{fossil} \times \delta_{i,fossil} \times X_{fossil}^b) + X_{fire}^b, \text{ i} = 1, 2, \dots, N \quad (4)$$

$$X_i^b = \left(X_{bio}^b + X_{ocn}^b + X_{fossil}^b + X_{fire}^b\right) + \lambda_{netflux} \times \delta_{i,netflux} \times (X_{bio}^b +$$

$$X_{ocn}^b + X_{fossil}^b + X_{fire}^b), \text{ i} = 1, 2, \dots, N \quad (5)$$




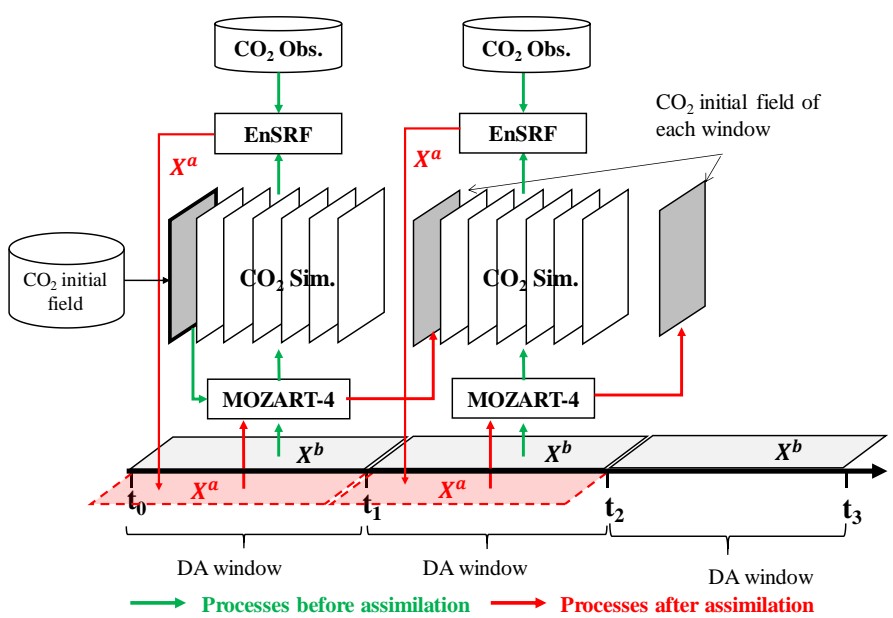


**Figure 1**. Flow chart of the GCASv2 system

### 2.1.1 EnSRF assimilation algorithm

To avoid storing and inverting very large matrices during analysis, the Ensemble
square root filter (EnSRF) algorithm, introduced by Whitaker and Hamill (2002), is
used to constrain the carbon fluxes in this version. EnSRF obviates the need to perturb
the observations in contrast to the traditional EnKF algorithm and assimilates
observations in a sequential way. It has a better performance than the method to
assimilate observations simultaneously as long as the observation errors are
uncorrelated (Houtekamer and Mitchell, 2001). The implementation process and setup
are detailed below.
After obtaining an ensemble of state vectors as described in Section 2.1, ensemble
runs of MOZART-4 are conducted to propagate these errors in the model with each
ensemble sample of a state vector. The background error covariance $\boldsymbol{P^b}$ is calculated
based on the forecast ensemble from Eq. (6):
$$\boldsymbol{P^b} = \frac{1}{n-1}\sum_{i=1}^{n}(\boldsymbol{X_i^b} - \overline{\boldsymbol{X}}^{\boldsymbol{b}})\,(\boldsymbol{X_i^b} - \overline{\boldsymbol{X}}^{\boldsymbol{b}})^T \qquad (6)$$





where $\overline{X^b}$ represents the mean of the ensemble samples. Based on the background
error covariance, the response of the uncertainty in the simulated concentrations to the
uncertainty in emissions is obtained. Combing observational vector $\boldsymbol{y}$, the state vector
is updated according to the following formulations:
$$\overline{X^a} = \overline{X^b} + \mathbf{K}(\mathbf{y} - H\overline{X^b}) \tag{7}$$

$$\mathbf{K} = \boldsymbol{P^b H^T}(\boldsymbol{H P^b H^T} + \boldsymbol{R})^{-1} \tag{8}$$

$$\delta X_i^a = \delta X_i^b - \widetilde{K}H\delta X_i^b \tag{9}$$

While employing sequential assimilation and independent observations
$$\widetilde{K} = (1 + \sqrt{R/_{HP^bH^T + R}})^{-1}\mathbf{K} \tag{10}$$

where $H$ is the observation operator that maps the state variable from model space to
observation space. $K$ is the Kalman gain matrix of ensemble mean depending on
background and observation error covariance $R$, representing the relative contributions
to analysis. $\widetilde{K}$ is the Kalman gain matrix of ensemble perturbation, and then emission
perturbations after inversion $\delta X_i^a$ can be calculated. At the analysis step, the ensemble
mean $\overline{X^a}$ is taken as the best estimate of the carbon flux.

To reduce the computational cost and the influence of representative errors, a

'super-observation' approach is adopted based on the optimal estimation theory
(Miyazaki et al., 2012). A super-observation is generated by averaging all observations
located within the same model grid within a DA window. We assume that the
observation errors of different stations at different times are independent of each other.
The standard deviation of the $j$th observation $y_j$ is $r_j$. The super-observation $y_{new}$,
standard deviation $r_{new}$ and corresponding simulations $x_{new,i}$ from one perturbed
prior flux $X_i^b$ are calculated:
$$1/_{r_{new}^2} = \Sigma_{j=1}^m 1/_{r_j^2} \tag{7}$$



$$y_{new} = \sum_{j=1}^{m} w_j \, y_j \big/ \sum_{j=1}^{m} w_j \qquad (8)$$

$$x_{new,i} = \sum_{j=1}^{m} w_j \, x_{j,i} \big/ \sum_{j=1}^{m} w_j \qquad (9)$$

where $w_j = \frac{1}{r_j^2}$ is the weighting factor; $m$ is the number of observations within a super-observation grid. The super-observation error decreases as the number of observations used for the super-observation increases.

**2.1.2 Atmospheric transport model**

Same as the GCAS system (Zhang et al., 2015), the global chemical transport Model for OZone And Related chemical Tracers (MOZART-4; Emmons et al., 2010) is adopted as the atmospheric transport model in GCASv2. MOZART-4 is a flexible model, it can be run at essentially any resolution, and can be driven by essentially any meteorological data set and with any emission inventories (Emmons et al., 2010). In this system, we preset two horizontal resolutions for MOZART runs, one being approximately 2.8°×2.8°, with model grids of 128 × 64, and another being approximately 1.0°×1.0°, with model grids of 360 × 180. In the vertical direction, we use 28 layers. The ERA-Interim reanalysis datasets from the European Centre for Medium-Range Weather Forecasts (ECMWF) are used to drive the model. ERA-Interim data set includes as many as 128 meteorological variables, and has the highest spatial resolution of approximately 80 km (T255 spectral) on 60 vertical levels from the surface up to 0.1 hPa. Only the variables required for MOZART-4 with a spatial resolution of 1.0°×1.0°, and 28 vertical levels for 3-D variables from the surface to approximately 2.5 hPa are selected in this system. The selected variables and vertical levels are shown in Table S1 and S2 in the supporting information.

**2.1.3 DA window and localization**

The DA window is set to one week in GCASv2, which is the same as before. Theoretically, a longer DA window is better, because $CO_2$ is a stable species. The longer window, the farther $CO_2$ will be transported. In this way, more observation stations will sense the flux change of one area, and thus more observations can be used to optimize





the flux of that place. However, the farther away, the weaker signal the stations can
sense. Limited by the method of EnKF, this weak signal will be masked by the method's
own unphysical signal (spurious correlation). In addition, Zhang et al. (2015) tested
different DA window lengths and found that the longer the window, the larger optimized
terrestrial carbon sink will be, resulting in a smaller optimized annual atmospheric $CO_2$
growth rate as compared to the observed rate. Therefore, they pointed out that the 1-
week DA window seems to be most suitable. For this reason, this study also uses the
same DA window of one week as before.

In the EnKF method, there are inevitably spurious correlations. Therefore, a

localization scale, which determines that only measurements located within a certain
distance (cutoff radius) from a grid point will influence the analysis of this grid, must
be set to reduce the effect of spurious correlations. The localization technique in this
study is based on both the distance between one site and one grid cell of λ, and the
linear correlation coefficient between the simulated concentrations and the perturbed
fluxes for each parameter (λ)/observation pair. If the distance is less than 500 km and
the correlation coefficient is greater than zero, the observations will be accepted for
assimilation, and if the distance is greater than/equal to 500 km and less than 3000 km
and the relationship between a parameter deviation and its modeled observational
impact is statistically significant ($p<0.05$), then that relationship is retained. Otherwise,
the relationship is assumed to be spurious noise. The scale of 3000 km is set simply
according to the globally-averaged 80-m wind speed during the day (4.96 m/s, Archer
and Jacobson, 2005) and the length of DA window (1 week).
**2.2 Prior carbon fluxes**

As described in Section 2.1, there are 4 types of prior carbon fluxes in GCASv2.

In this study, FOSSIL carbon emissions are obtained from NOAA's CarbonTracker,
version    CT2017    (Peters    et    al.    2007,    with    updates    documented    at
http://carbontracker.noaa.gov), which is an average of the Carbon Dioxide Information
Analysis Center (CDIAC) product (Andres et al., 2011) and the Open-source Data
Inventory of Anthropogenic $CO_2$ (ODIAC) emission product (Oda and Maksyutov,





2011). The FIRE $CO_2$ emissions are also taken from CT2017, which are the average of
the Global Fire Emissions Database version 4.1 (GFEDv4) (van der Werf et al., 2010;
Giglio et al., 2013) and the Global Fire Emission Database from the NASA Carbon
Monitoring System (GFED_CMS). The OCN $CO_2$ exchange is from the $pCO_2$-Clim
prior of CT2017, which is derived from the Takahashi et al. (2009) climatology of
seawater $pCO_2$. In addition, as shown in Figure 7 of the CarbonTracker Documentation
CT2017   release   (https://www.esrl.noaa.gov/gmd/ccgg/carbontracker/CT2017/,
accessed on 4 Mar, 2020), there are no data in many seas like Japan Sea, Mediterranean,
Gulf of Mexico, East China Sea, and so on, and therefore, the fluxes in 2009 modeled
using the global ocean circulation (OPA) and the biogeochemistry model (PISCES–T)
(Buitenhuis et al., 2006; Jiang et al., 2013) is used to fill the no data areas.
The BIO carbon flux, which is the most important prior carbon flux in an
assimilation system, was simulated using the Boreal Ecosystems Productivity
Simulator (BEPS) model (Chen et al., 1999; Ju et al., 2006) in this study. BEPS is a
process-based, remote sensing data driven, and mechanistic ecosystem model. In this
study, BEPS model was run starting from 2000. To simplify the initialization, the initial
values of the different carbon pools are from a previous BEPS simulation (Chen et al.,
2019). In short, all carbon pools were assumed to be in a state of dynamic equilibrium
from 1901 to 1910. And all carbon pools were determined by solving a set of equations
describing the dynamics of carbon pools (Chen et al., 2003). Then the simulation
forwarded using historical data. Due to the lack of historical data of remote sensed LAI
data, the averaged LAI from 1982 to 1986 represented that over the 1901-1981 period.
Then, all our initial carbon pools were set to states of carbon pools in 2000 according
to Chen et al. (2019). The BEPS model was also driven by the 1°×1° ERA-Interim
reanalysis datasets, including relative humidity, wind speed, air temperature, incoming
solar radiation, and total precipitation. The other data include LAI data and clumping
index. LAI was inverted from surface reflectance datasets of Moderate Resolution
Imaging Spectroradiometer (MODIS) (Liu et al., 2012), and the clumping index was
derived from the MODIS Bidirectional Reflectance Distribution Function (BRDF)
products, which provided the finest pseudo multi-angular data for the land surface,
according to Normalized Difference between Hotspot and Darkspot (NDHD) (Chen et
al., 2005, He et al., 2012).
**2.3 GOSAT XCO$_2$ retrievals**
The GOSAT XCO$_2$ retrievals of the ACOS Version 7.3 Level 2 Lite product
(O'Dell et al., 2012; Crisp et al., 2012) at the pixel level during May 2009 ~ Dec 2015
is used in this study, which is bias-corrected (Wunch et al., 2011). In order to achieve
the most extensive spatial coverage with the assurance of using best quality data
available, before being used in the inversion system, the XCO$_2$ retrievals are filtered
with two parameters of warn_levels and xco2_quality_flag, which are provided along
with the product. Only the data with xco2_quality_flag greater than 0 are selected. The
selected data are then divided into three groups according the value of warn_levels, that
are with warn_levels less than 8, greater than 9 and less than 12, and greater than 13,
respectively. The group with smallest warn_levels has the best data quality, while that
with the largest is the worst. Then, the pixel data are averaged within the grid cell of
1°×1°, and in each grid, only the group with best data quality is selected and then
averaged. The other variables like column-averaging kernel, retrieval error and so on
which are provided along with the XCO$_2$ product are also dealt with the same method.
This process is the same as Wang et al. (2019).
For the modeled XCO$_2$, the simulated CO$_2$ concentration profile should be first
mapped into the satellite retrieval levels and then vertically integrated according to the
following equation.

$$XCO_2^m = XCO_2^a + \sum_j h_j a_j (A(x) - y_{a,j}) \tag{10}$$


where $j$ denotes the retrieval level; $x$ is the simulated CO$_2$ profile, and $A(x)$ is a mapping
matrix; XCO$_2^a$ is the prior XCO$_2$; $h_j$ is a pressure weighting function, $a_j$ and $y_a$ are the
satellite column averaging kernel and the prior CO$_2$ profile for retrieval, respectively.
Except the simulated CO$_2$ profile, the other quantities are provided along with the
ACOS product and filtered and averaged to 1°×1° grid according to the above method.


**2.4 Evaluation data and method**


Generally, direct validation of the optimized flux is impossible, and instead, we
indirectly evaluate the posterior flux by comparing the forward simulated atmospheric
$CO_2$ mixing ratios against measurements (e.g., Jin et al., 2018; Wang et al., 2019; Feng
et al., 2020). First, the simulated $XCO_2$ are compared against the corresponding GOSAT
$XCO_2$ retrievals to test the effectiveness of the assimilation system (see Section 2.3 for
the description of the GOSAT $XCO_2$ retrieval). Second, Surface $CO_2$ observations used
for independent evaluations in this study are obtained from the
obspack_co2_1_GLOBALVIEWplus_v5.0_2019-08-12 product. It is a subset of the
Observation Package (ObsPack) Data Product (ObsPack, 2019), and contains a
collection of discrete and quasi-continuous measurements at surface, tower and ship
sites contributed by national and universities laboratories around the world. In this study,
surface $CO_2$ measurements from 52 flask sites are selected to evaluate the posterior $CO_2$
concentrations, which are all provided by the NOAA Global Monitoring Laboratory
(with lab number of 1 in each filename). The locations of the 52 sites could be found in
Figure 2 and the corresponding sites code as well as the information latitude and
longitude are listed in Table S3 in the Supporting Information.
During the evaluation, 3 basic statistical measures, namely mean bias (BIAS), root
mean square error (RMSE), and correlation coefficient (CORR), are calculated against
the surface $CO_2$ observations and GOSAT $XCO_2$ retrievals, respectively. The BIAS,
RMSE, and CORR reflect the overall model tendency, both the model bias and error
variance, and the linear correspondence between the modeled and observational
values/retrievals, respectively. The functions of these 3 basic statistical measures are
expressed as:
$$BIAS = \frac{1}{M}\sum_{j=1}^{M}(x_j - y_j) = \bar{y} - \bar{x} \qquad (10)$$

$$RMSE = \sqrt{\frac{1}{M}\sum_{j=1}^{M}(x_j - y_j)^2} \qquad (11)$$



$$CORR = \frac{\sum_{j=1}^{M}(x_j - \bar{x})(y_j - \bar{y})}{\sqrt{\sum_{j=1}^{M}(x_j - \bar{x})^2}\sqrt{\sum_{j=1}^{M}(y_j - \bar{y})^2}}$$
(12)

where $x_j$ and $y_j$ denote the modeled and the observational values/retrievals,
respectively, at the $j$th out of $M$ records, and the overbars denote averages.

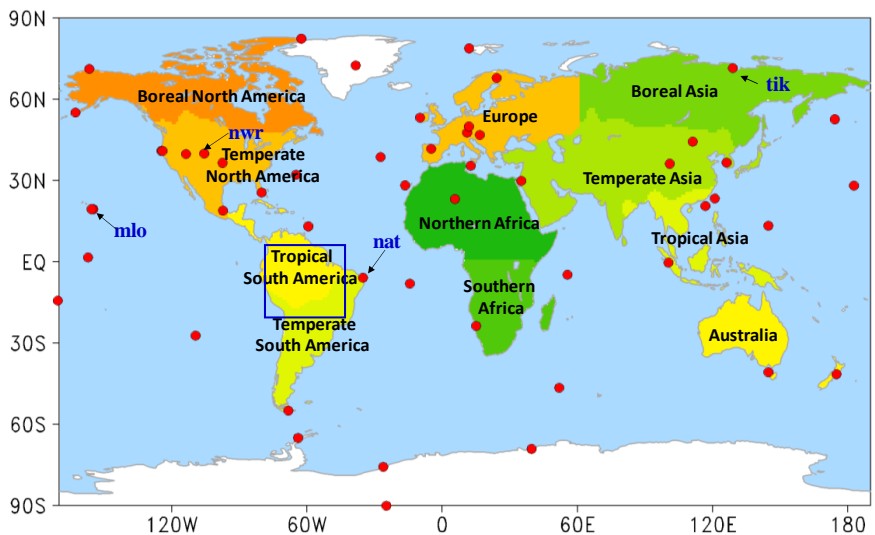

**Figure 2**. Distributions of the observation sites used in this study. Red solid circles are
the 52 surface flask sites used for evaluations, the shaded shows the 11 TRANSCOM
regions, the blue rectangle shows the Amazon region, which is defined the same as
Botta et al. (2012)

## 3. Experimental Design

The assimilation system was run from May 1, 2009 to Dec 31, 2015. Two forward
simulations with the prior and posterior fluxes were also conducted from May 1, 2009
to Dec 31, 2015, respectively. For both assimilation and forward runs, the initial field
of 3-D $CO_2$ concentrations at 00:00 UTC May 1, 2009 was from the product of CT2017
as well, and the MOZART-4 model was run with the resolution of 2.8°×2.8°. The first
8 months are considered as a spin-up run, and the results from Jan 1, 2010 to Dec 31,
2015 are analyzed and evaluated in this study.
During the assimilation, the resolution of λ is the same as the transport model. The





BIO $CO_2$ exchanges and OCN fluxes are optimized in this study, and the FOSSIL and
FIRE carbon emissions are kept intact. Following Wang et al., (2019), global annual
uncertainties of 100% and 40% are assigned to BIO and OCN $CO_2$ exchanges,
respectively. Accordingly, the uncertainties of the scaling factor ($\lambda$) for the prior BIO
and OCN fluxes in each DA window at the grid cell level are assigned to 3 and 5,
respectively. The model-data mismatch error of $XCO_2$ is constructed using the GOSAT
retrieval error, which is provided along with the ACOS product. According to the
previous works of Wang et al. (2019) and Deng et al. (2014), all retrieval errors are also
uniformly inflated by a factor of 1.9 in this study, which is the same as Wang et al.
(2019), but a lowest error is added in this study, which is fixed as 1 ppm.
**4. Results and Discussions**
**4.1 Evaluation for the inversion results**
**4.1.1 Evaluation using Assimilated GOSAT $XCO_2$ retrievals**
Figure 3a shows the zonal mean $XCO_2$ model-data mismatch errors at different
latitudes during the study period. Compared with the GOSAT $XCO_2$ retrievals, basically
all the zonal mean BIAS of the prior $XCO_2$ in different latitudes are greater than 1 ppm,
with a global mean of 1.8±1.3 ppm (average ± standard deviation, same thereafter), but
for the posterior $XCO_2$, most zonal average BIAS are within ±0.5 ppm, with global
mean of -0.0±1.1 ppm. The global mean RMSE between the simulated and GOSAT
retrieved $XCO_2$ concentrations also decreases from a prior value of 2.2 ppm to 1.1 ppm
(Table 1), indicating that the model-data mismatch errors between the simulated and
retrieved $XCO_2$ are significantly reduced. Overall, for both prior and posterior
concentrations, the BIAS in the southern hemisphere is smaller than that in the northern
hemisphere. In the same hemisphere, the BIAS at low latitudes is smaller than that at
high latitudes. Figure 4 shows the spatial distribution of the posterior $XCO_2$ biases. It
could be found that in most grids (~80%), the biases are within ±1ppm. In Tropical
Pacific, North Pacific, North Atlantic and Tropical Land, most biases are positive, and



in the northern extra-tropical lands, negative biases are dominant. This pattern may be
related to the retrieval errors, and the large BIAS in the high latitudes may be attributed
to the large retrieval errors in those areas, which are caused by the lower solar elevation
angle. Overall, this small posterior BIAS, which is less than the retrieval error (Crisp et
al., 2012), indicates that the GCAS system works well with the GOSAT $XCO_2$ retrievals
in this study.

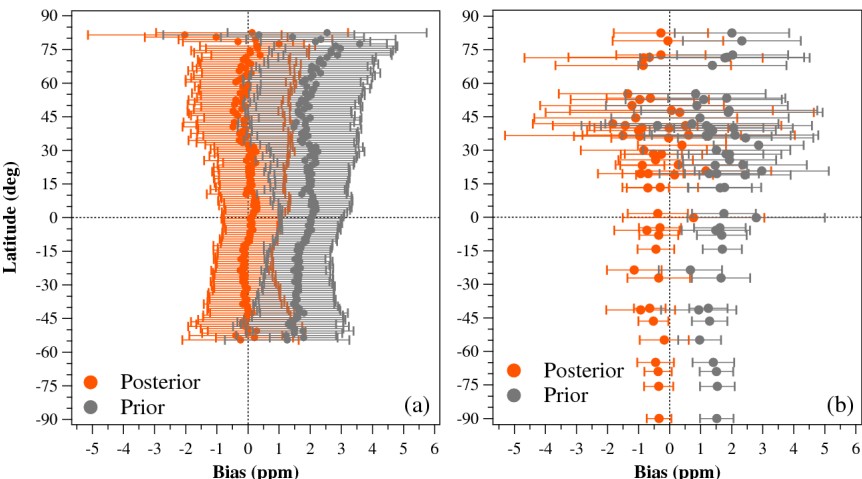


**Figure 3.** Biases at different latitudes (a, simulated and retrieved $XCO_2$; b, simulated
and observed $CO_2$ mixing ratios; error bar represents the standard deviations of the
biases at each latitude and each site, respectively)

**Table1.** Statistics of the simulated surface $CO_2$ and $XCO_2$ concentrations against the
surface flask observations and GOSAT retrievals, respectively

| | BIAS (ppm)* | | RMSE (ppm) | | CORR | |
|---|---|---|---|---|---|---|
| | Prior | Posterior | Prior | Posterior | Prior | Posterior |
| $XCO_2$ | 1.8±1.3 | -0.0±1.1 | 2.2 | 1.1 | 0.95 | 0.96 |
| Surface $CO_2$ | 1.6±1.8 | -0.5±1.8 | 2.4 | 1.9 | 0.96 | 0.96 |

*mean ± standard deviation

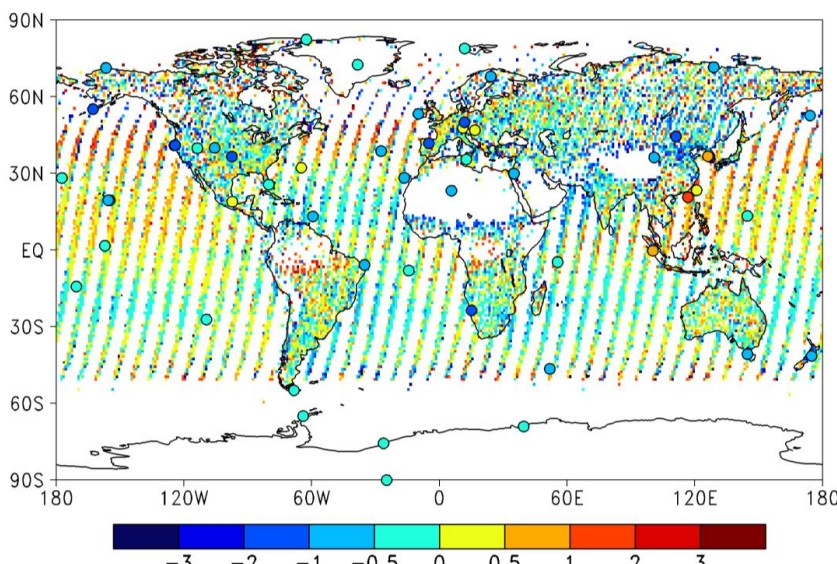


**Figure 4.** Distributions of the mean biases of the posterior (cycle) surface $CO_2$ and

(grid shaded) $XCO_2$ concentrations (simulations minus observations/retrievals)



**4.1.2 Evaluation using independent surface observations**


Figure 3b shows the mean biases of the simulated surface $CO_2$ mixing ratios at each


flask site at different latitudes. It could be found that the BIAS of the prior $CO_2$ mixing


ratios are basically greater than 1 ppm at different latitudes, with global mean of 1.6±1.8


ppm, after constraining using the GOSAT $XCO_2$ retrievals, the BIAS at most sites are


within ±1 ppm, with a global mean of -0.5±1.8 ppm. These BIAS are similar to those


of Basu et al. (2013), in which the average model–observation bias decreased from a


prior value of 1.95 ppm to -0.55 ppm. In our study, the RMSE between the simulated


and surface flask concentrations are also reduced in most sites, with the global mean


RMSE decreasing from 2.4 to 1.9 ppm (Table 1). The BIAS in the northern hemisphere


are significantly larger than those in southern hemisphere, because the carbon flux in


the northern hemisphere is more complex than that of the southern hemisphere (Wang


et al., 2019). In addition, the posterior BIAS in most sites are negative, especially in the


middle latitudes in the northern hemisphere. The significant negative biases (less than


1 ppm) are mainly distributed in North America, Europe, central Asia, while positive






biases are mainly located along east Asian coast (Figure 4), indicating that the carbon
sinks in North America and Europe might be overestimated in this study, while those in
the upwind areas of east Asian coastal sites, mainly eastern China, may be
underestimated.
Moreover, it also could be found that the global mean prior BIAS of $XCO_2$ (about
1.82 ppm) is greater than the surface concentrations (1.60 ppm), while the BIAS of
$XCO_2$ reduced by inversion (about 1.8 ppm) is less than the reduction of BIAS in the
surface concentrations (about 2.1 ppm). This may be attributed to the fact that, on the
one hand, although the GOSAT $XCO_2$ retrievals were bias-corrected, there may still be
some systematic deviations; on the other hand, the responses of surface observations to
changes in the surface carbon flux is faster than the $XCO_2$ concentrations, so that larger
flux adjustments are needed to match $XCO_2$ concentration with ground data. A similar
situation was reported in Wang et al. (2019). In their study, GOSAT $XCO_2$ retrievals
were used to optimize the terrestrial carbon flux in 2015. Their inversion reduced the
BIAS of simulated surface and $XCO_2$ (compared against TCCON sites) concentrations
by about 1.1 and 0.9 ppm, respectively.
Figure 5 shows the time series of simulated and observed $CO_2$ mixing ratios at four
sites, i.e., mlo, nwr, tik, and nat. The mlo and nwr sites are two mountain stations located
in the center of Pacific and western US, respectively, and nat and tik are two coastal
sites located in Amazon and Siberia, respectively (Figure 2). Overall, the posterior
mixing ratios have a better agreement with the observations at all 4 sites. The mlo site
is an atmospheric baseline station. At mlo, the posterior mixing ratio well reproduces
the observed concentration, while the prior concentrations are overestimated all the
time since the summer of 2010, especially during the summertime every year. Besides,
the posterior concentrations during the wintertime are underestimated, and the
underestimation gradually increases along with time. A similar situation also could be
found at the nat site as well as other sites located in tropical and southern hemisphere
oceans (Figure not shown). Figure S1 shows the interannual variations of the global
mean BIAS. Clearly, the biases of surface $CO_2$ are gradually accumulated, leading to
the relatively large mean bias (-0.5 ppm). If we remove the impact of accumulation, the
annual bias is about -0.1 ppm per year (about -0.2 PgC yr$^{-1}$). There are no error
accumulations at most land sites like nwr and tik. These indicate that the global net
carbon sinks are slightly overestimated every year, but in different lands, there are
interannual variations.

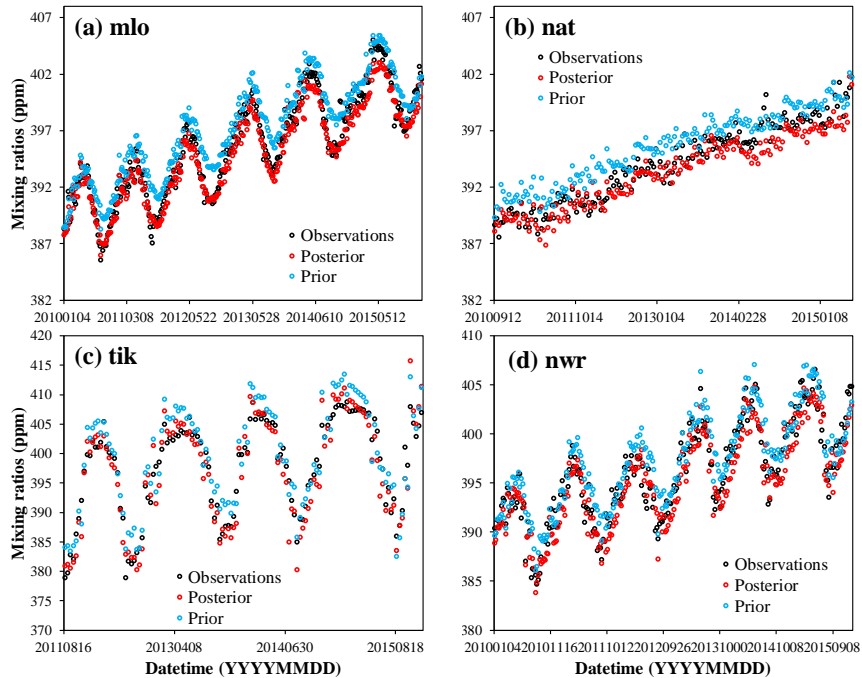


**Figure 5.** Modeled and observed $CO_2$ time series at four surface stations
**4.2 Global Carbon Budget**

Table 2 presents the mean prior and posterior global carbon budgets during 2010 ~

2015 of this study. For comparison, the mean global carbon budgets from Global
Carbon Budget 2018 (GCP2018, Le Quéré et al., 2018), CT2017, and Jena CarboScope
(JCS, Rödenbeck, 2005) are also shown. Both CT2017 and JCS estimates of the
surface-atmosphere $CO_2$ exchange were based on the atmospheric measurements of
$CO_2$ concentrations. In this study, the JCS product of s04oc_v4.3 is adopted. It should
to be noted that JCS only provides the land-atmosphere carbon flux, which is the sum





of BIO carbon flux and FIRE carbon emissions, and no individual FIRE carbon
emissions data is available. To compare, the FIRE carbon emissions used in this study,
which is from CT2017, is also applied to the JCS data, namely the BIO carbon flux of
JCS in this manuscript is obtained from the land-atmosphere carbon flux of JCS minus
the FIRE carbon emission of this study.
**Table 2**. Mean global carbon budgets during 2010 ~2015 estimated in this study as well
as those from the prior fluxes, GCP2018, CT2017 and JCS (PgC yr$^{-1}$)

|  | Prior | Posterior | GCP2018 | CT2017 | JCS |
|---|---|---|---|---|---|
| Fossil fuel and industry (FOSSIL) | 9.58 | 9.58 | 9.49 | 9.62 | 9.31 |
| Biomass burning (FIRE) | 2.02 | 2.02 | 1.52* | 2.03 | 2.02 |
| Terrestrial ecosystem (BIO) | -4.07 | -4.24 | -3.13 | -4.29 | -4.07 |
| Ocean (OCN) | -2.47 | -2.56 | -2.46 | -2.57 | -2.25 |
| Budget imbalance | - | - | -0.52 | - | - |
| Net biosphere exchange*** | -2.05 | -2.22 | -2.12 | -2.27 | -2.05 |
| Global net carbon flux (AGR) | 5.06 | 4.80 | 4.91** | 4.79 | 5.01 |

* land-use change emissions, **atmospheric growth in GCP2018, *** for GCP2018, it
is the sum of BIO, FIRE and budget imbalance, and for the others, it is the sum of BIO
flux and FIRE emission.
The mean posterior BIO carbon flux during 2010-2015 in this study is -4.24 PgC
yr$^{-1}$ (negative/positive mean carbon uptake/release from/to the atmosphere, same
thereafter), and the OCN flux is -2.56 PgC yr$^{-1}$, after considering the FOSSIL carbon
emission (9.58 PgC yr$^{-1}$) and FIRE carbon emission (2.02 PgC yr$^{-1}$), the mean global
net carbon flux (i.e., atmospheric $CO_2$ growth rate) inverted in this study is 4.80 PgC
yr$^{-1}$. Both the posterior BIO and OCN carbon fluxes are stronger than the prior ones,
and the posterior global net carbon flux is weaker than the prior one. Compared with
the others, both posterior BIO and OCN fluxes are close to the ones of CT2017, but
higher than the ones of JCS. The atmospheric $CO_2$ growth rate (AGR) of GCP2018 was
estimated directly from atmospheric $CO_2$ measurements, which were provided by the
US National Oceanic and Atmospheric Administration Earth System Research
Laboratory (NOAA/ESRL) (Dlugokencky and Tans, 2018), and therefore, it could be
considered as a true value. The posterior AGR in this study (4.8 PgC yr$^{-1}$) is slightly





lower than GCP2018 and very close to CT2017. Compared with GCP2018, the
deviations of prior and JCS AGR are 0.15 and 0.10 PgC yr$^{-1}$, while the ones of posterior
and CT2017 are -0.11 and -0.12 PgC yr$^{-1}$, respectively.

**4.3 Regional Carbon Flux**

Figure 6 shows the distributions of the mean prior and posterior annual BIO and
OCN carbon fluxes as well as their differences during 2010 - 2015. For the prior BIO
flux, carbon uptakes mainly occur over eastern North America, Amazon, southern
Brazil, western Europe, southern Russia, eastern China, South Asia and Malay
Archipelago; and carbon releases mainly occur in western North America, eastern
Amazon, Argentina, most Africa, Indo-China Peninsula, and parts of eastern Europe
and Russia. For the prior OCN flux, carbon uptakes mainly happen in mid-latitude
regions in both hemispheres, while carbon sources are mainly in tropical oceans and
Southern Ocean. After the constraint with the GOSAT XCO$_2$ retrievals, the overall
patterns of carbon sinks and sources are similar to the prior ones. However, the BIO
sinks in East and Central America, eastern Amazon, tropical Africa, Indo-China
Peninsula, and southwestern Russia are obviously increased, on the contrary, in western
North America, temperate South America, extra-tropical Africa, South Asia, Southwest
China, North China, Siberia, and parts of southern and northern Europe, the carbon
sources are increased. For the OCN flux, in most tropical and northern hemisphere
oceans, the carbon sinks are slightly increased, while in most southern hemisphere
oceans, the carbon sources are slightly enhanced.

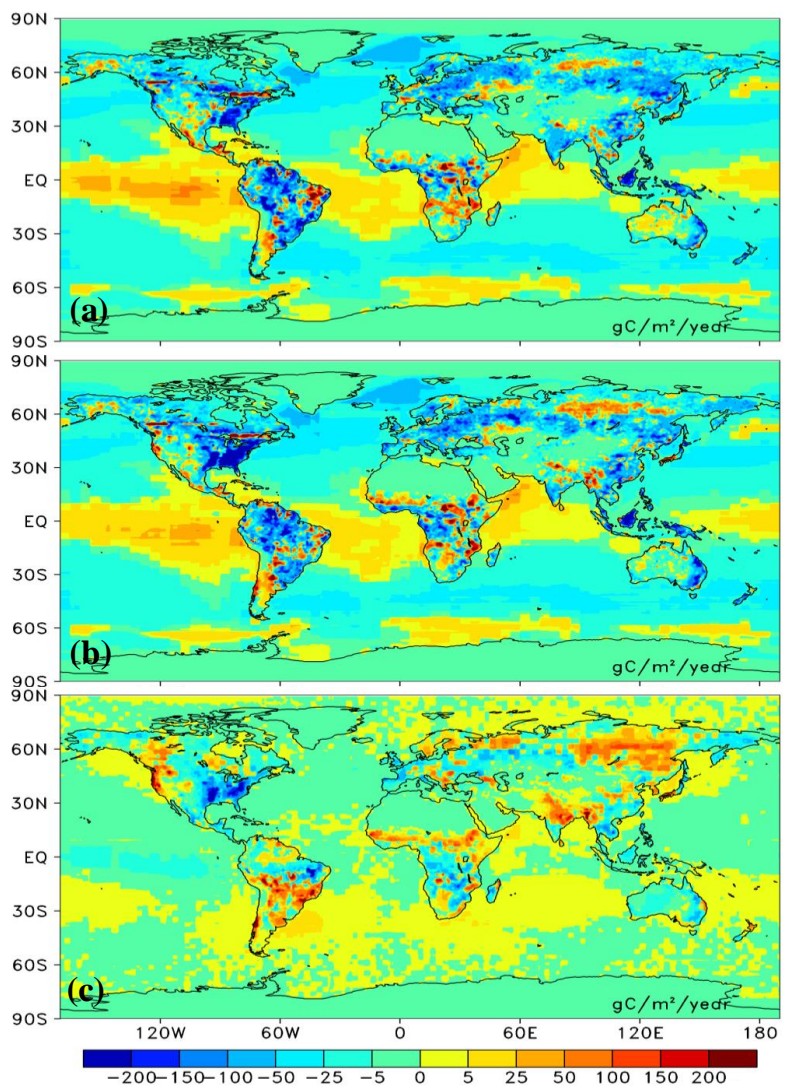


**Figure 6.** Distributions of mean annual terrestrial ecosystem and ocean carbon fluxes


a) prior and b) posterior and c) their differences (posterior - prior) (gC m$^{-2}$yr$^{-1}$)

Table 3 lists the aggregated mean annual prior and posterior BIO carbon fluxes

during 2010-2015 for the 11 TRANSCOM land regions (Figure 2, Gurney et al., 2002)
as well as 3 aggregated large-scale regions, i.e., Northern Lands, Tropical Lands, and
Southern Lands. Northern lands include Boreal North America, Temperate North
America, Boreal Asia, Temperate Asia and Europe; Tropical Lands include Tropical





515 South America, Tropical Asia, Northern Africa and Southern Africa; and Southern

516 Lands include Temperate South America and Australia. For the prior, there is a largest

517 carbon sink in Tropical South America, followed by Boreal Asia and Temperate Asia,

518 and a weakest carbon flux in Southern Africa. After optimization using GOSAT $XCO_2$

519 retrievals, the carbon sinks of Temperate North America, Southern Africa are

520 significantly increased, and those in Australia and Europe are also enhanced. However,

521 in Temperate South America, Northern Africa, Boreal Asia, and Temperate Asia, the

522 carbon sinks are decreased. Very small changes are found in Boreal North America,

523 Tropical South America, and Tropical Asia, especially for Tropical South America,

524 however, as shown in Figure 6, there are obvious changes over different areas in

525 Tropical South America, thus the zero change in statistics in this region may be just a

526 coincidence. For the Amazon region (Figure 2), the estimated BIO flux is decreased

527 from a prior of -0.52 PgC yr$^{-1}$ to -0.45 PgC yr$^{-1}$. The largest carbon sink occurs in

528 Temperate North America, followed by Tropical South America and Europe, and the

529 weakest sink appears in Northern Africa.

530  For comparisons, Table 3 also lists the mean BIO carbon fluxes of CT2017 and

531 JCS for the same period. For the 3 large-scale regions, i.e., Northern Lands, Tropical

532 Lands and Southern Lands, the same as the global total BIO carbon sink, the carbon

533 sinks in these 3 regions are also similar to CT2017. However, in each region, the

534 distributions of carbon sinks between this study and CT2017 are significantly different.

535 In Northern Lands, the carbon sinks estimated by this study are more evenly distributed,

536 although Temperate North America has the largest carbon sink, and those in Boreal Asia,

537 Temperate Asia and Europe are also very strong and comparable. However, in CT2017,

538 the carbon sinks are mainly distributed in Boreal Asia and Temperate Asia, accounting

539 for more than 70% of the total sink in Northern Lands. The sinks in Temperate North

540 America and Europe are very weak or even neutral. In Tropical Lands, this study shows

541 strong carbon sinks in Tropical South America and Tropical Asia, and a weak sink in

542 Africa, while CT2017 shows an opposite pattern. In Southern Lands, this study shows

543 comparable sinks in Temperate South America and Australia, while CT2017 shows a





strong sink in Temperate South America and very weak one in Australia. Compared
with JCS, except for Temperate North America and Southern Africa, the carbon sinks
are comparable in other regions. Constraining with different observations might be one
of the main reasons among these studies. Many studies have shown differences between
the constraints with in situ observations and $XCO_2$ retrievals (e.g., Wang et al., 2019;
Deng et al., 2014). Besides, these differences may be also related to the different prior
BIO carbon fluxes among these studies, especially for the tropical land. The distribution
of the posterior BIO fluxes in this study and CT2017 are consistent with the
corresponding prior fluxes in the tropical land (Table 3). Using the same GOSAT $XCO_2$
retrievals, Deng et al. (2014) adopted a similar prior flux with this study, which was
also simulated using the BEPS model but globally neutralized, to infer the land fluxes
of 2010, their distributions are roughly consistent with this study, while Wang et al.
(2019) applied the prior flux from CT2016 to optimizing the fluxes in 2015, and they
showed a similar distribution of land sinks over tropical lands to that of CT2017.
**Table 3**. Regional BIO and FIRE flux in the 11 TRANSCOM land regions (PgC yr$^{-1}$)

| Regions | Fire | This Study | | CT2017 | | JCS |
|---|---|---|---|---|---|---|
| | | Prior | Posterior | Prior | Posterior | |
| Boreal North America | 0.065 | -0.26 | -0.28 | -0.05 | -0.39 | -0.31 |
| Temperate North America | 0.022 | -0.49 | -0.88 | -0.14 | -0.23 | -0.21 |
| Tropical South America | 0.220 | -0.66 | -0.66 | 0.02 | -0.11 | -0.43 |
| Temperate South America | 0.142 | -0.3 | -0.15 | -0.16 | -0.42 | 0.13 |
| Northern Africa | 0.385 | -0.18 | -0.05 | -0.47 | -0.82 | -0.11 |
| Southern Africa | 0.628 | 0.01 | -0.14 | -0.63 | -0.55 | -0.66 |
| Boreal Asia | 0.097 | -0.61 | -0.45 | -0.18 | -0.99 | -0.51 |
| Temperate Asia | 0.065 | -0.51 | -0.42 | -0.15 | -0.66 | -0.69 |
| Tropical Asia | 0.258 | -0.45 | -0.47 | -0.05 | -0.07 | -0.73 |
| Australia | 0.097 | -0.16 | -0.23 | -0.15 | -0.07 | -0.08 |
| Europe | 0.015 | -0.46 | -0.52 | -0.18 | 0 | -0.44 |
| Northern Lands* | 0.26 | -2.33 | -2.55 | -0.7 | -2.27 | -2.16 |
| Tropical Lands** | 1.49 | -1.28 | -1.32 | -1.13 | -1.55 | -1.93 |
| Southern Lands*** | 0.24 | -0.46 | -0.38 | -0.31 | -0.49 | 0.05 |

*Northern lands include Boreal North America, Temperate North America, Boreal Asia, Temperate
Asia and Europe; **Tropical Lands include Tropical South America, Tropical Asia, Northern Africa
and Southern Africa; ***Southern Lands include Temperate South America and Australia.





Compared with other studies, the land fluxes (including FIRE but excluding
FOSSIL) in South America (-0.45 PgC yr$^{-1}$), Europe (-0.51 PgC yr$^{-1}$), Boreal Asia (-
0.35 PgC yr$^{-1}$), Temperate Asia (-0.35 PgC yr$^{-1}$), Tropical Asia (-0.21 PgC yr$^{-1}$), and
Australia (-0.13 PgC yr$^{-1}$) are comparable with the forest sinks in these regions during
2000-2007 estimated using forest inventory data by Pan et al. (2011). However, the land
fluxes in Africa and North America are significantly different from the estimates of Pan
et al. (2011). In North America, based on inventory-based calculations, the Second State
of the Carbon Cycle Report (SOCCR2, Hayes et al., 2018) estimated that the average
annual net land ecosystem flux was -0.96 PgC yr$^{-1}$, and after considering the outgassing
and wood products emissions, they reported the land-based carbon sink was -0.606 PgC
yr$^{-1}$ (±75%) during the 2004 to 2013 time period. The land flux estimated in this study
(-1.07 PgC yr$^{-1}$) is close to the bottom-up estimate of the net land ecosystem flux, but
much stronger than the reported land-based carbon sink of SOCCR2. In Africa, Ciais
et al. (2011) shown a comprehensive estimate for its carbon balance, given a sink of -
0.2 PgC yr$^{-1}$ (excluding land-use change emissions) based upon observations. Our
estimate of the BIO flux in Africa is very consistent with this result. Moreover, most
recently, Palmer et al. (2019) inferred the carbon fluxes of pan-tropical lands in 2015
and 2016 using both GOSAT and the NASA Orbiting Carbon Observatory (OCO-2)
$XCO_2$ retrievals, and their estimated net carbon emissions from African biosphere
dominate pan-tropical atmospheric $CO_2$ signals are similar to the results of this study.
In Boreal Asia, the land sink estimated by bottom-up approaches was in the range of -
0.11 ~ -0.76 PgC yr$^{-1}$ (Hayes et al., 2011; Nilsson et al., 2003; Dolman et al., 2012;
Zamolodchikov et al., 2017). CarbonTracker usually reports a very stronger carbon sink
(Jacobson et al. 2020; Peter et al., 2007; Zhang et al., 2014), one possible reason is that
there are no enough surface observations in Asia boreal regions. Saeki et al. (2013b)
conducted an inversion with a focus on the Siberia region, and also derived a large sink
of $-0.56 \pm 0.79$ PgC yr$^{-1}$ only using the NOAA data, but after adding additional
observations in Siberia, they obtained a weaker uptake of $-0.35 \pm 0.61$ PgC yr$^{-1}$. Our
estimate (-0.35 PgC yr$^{-1}$) is in the range of bottom-up estimates, and very consistent



with the Siberia-focused inversion (Saeki et al., 2013b). In Europe, previous GOSAT-
based inversions consistently derived a very large European sink, which was in the
range of -0.6 ~ -1.8 PgC yr$^{-1}$(Basu et al., 2013, Chevallier et al., 2014; Deng et al.,
2014), while the ones constrained using surface observations were much weak, in the
range of 0 ~ -0.4 PgC yr$^{-1}$ (Peters et al., 2007, 2010; Peylin et al., 2013; Scholze et al.,
2019). Our estimate of the BIO flux in Europe is smaller than the previous GOSAT-
based inversions, and close to the estimate of Pelylin et al. (2013). In the Amazon region,
the posterior land flux is -0.45 PgC yr$^{-1}$, which is in the range of the previous long-term
forest biomass sink estimates of -0.28 ~ -0.49 PgC yr$^{-1}$ (Phillips et al., 2009; Brienen et
al., 2015), but larger than the other inversions (e.g., Deng et al., 2016; Gatti et al., 2014).
**4.4 Interannual variations**
**4.4.1 Global land and ocean fluxes**

Figure 7 shows the interannual variations of the prior and posterior BIO and OCN

fluxes. Overall, from 2010 to 2015, the prior BIO fluxes show an increasing trend, but
for the posterior fluxes, there is no significant trend. Large differences between the prior
and the posterior fluxes mainly occur in 2010 and 2015. In 2010, the posterior sink is
much stronger than the prior, while in 2015, the posterior sink is much weaker than the
prior. For the OCN flux, both prior and posterior fluxes show consistently upward
trends, and except for 2015, the posterior sinks are basically stronger than the prior ones
every year. For the AGR (Figure 8), the prior sink shows a significant downward trend,
while the posterior one shows a slightly increasing trend. The same as the BIO fluxes,
large differences mainly occur in 2010 and 2015.





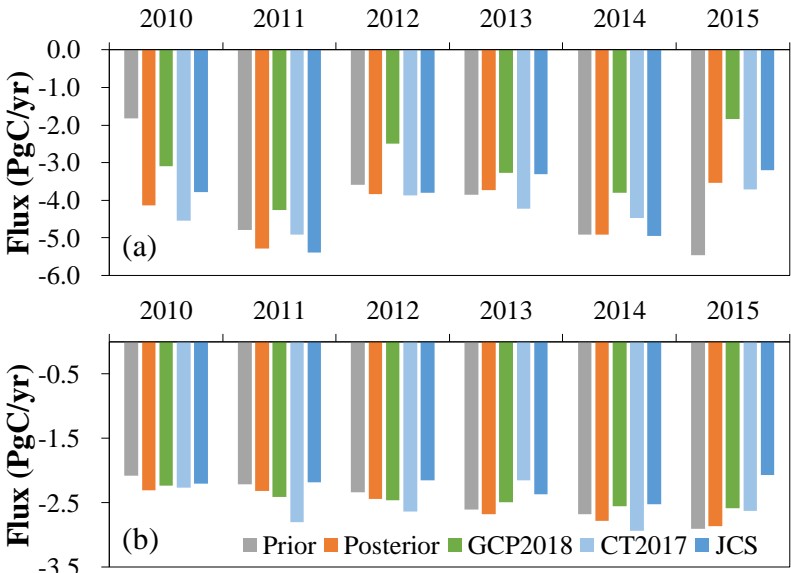

**Figure 7**. Interannual variations of global (a) BIO and (b) OCN fluxes of the prior and

posterior as well as GCP2018, CarbonTracker 2017 (CT2017) and Jena CarboScope

(JCS)

Compared with the other products, the interannual variations of the posterior BIO

fluxes (Figure 7a) are consistent with the inversions of CT2017 and JCS, and the

estimates of GCP2018. For each year, the inversions of this study are all in the range of

CT2017 and JCS, but higher than GCP2018. However, because GCP2018 has the item

of budget imbalance and the land-use change emission is different from the FIRE

emission, the BIO flux in GCP2018 is different from this study, so direct comparison

with GCP2018 is not meaningful. For OCN fluxes, overall, there are no significant

differences among different estimates, and the upward trend of this study is similar to

that of GCP2018, and higher than those of CT2017 and JCS. The interannual variation

of AGR in this study is also very consistent with GCP2018 (Figure 8). Except for 2012

and 2015, the absolute deviations of AGR between this study and GCP2018 are within

0.3 PgC yr$^{-1}$.





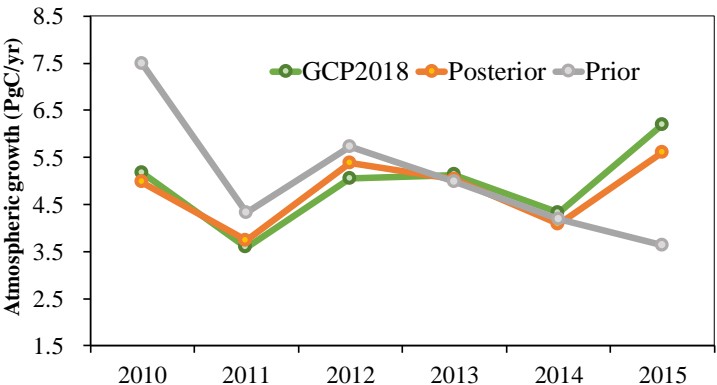


**Figure 8**. Interannual variations of the atmospheric $CO_2$ growth rates

**4.4.2 Regional land fluxes**
Figure 9a, b, and c show the prior and posterior interannual variations of the BIO
fluxes in Northern Lands, Tropical Lands and Southern Lands, respectively. In Northern
Lands, the interannual variations of both prior and posterior fluxes are similar to the
corresponding global land totals (Figure 7a), i.e., upward trend for the prior flux and no
trend with the posterior one, indicating that the interannual variations of global BIO
fluxes are dominated by the fluxes in Northern Lands. In Tropical Lands, the
interannual variations of posterior fluxes are similar to the prior ones, however,
compared with the prior sinks in 2010 and 2011, the posterior sinks are much stronger,
while in 2013 and 2015, they are much weaker. In Southern Lands, there are large
differences for the interannual variations between the prior and posterior fluxes. For the
prior flux, the highest sink is in 2011 and the weakest in 2012, and after that, it increases
year by year, while for the posterior flux, the sink decreases from 2010 to 2013, and
then increases.





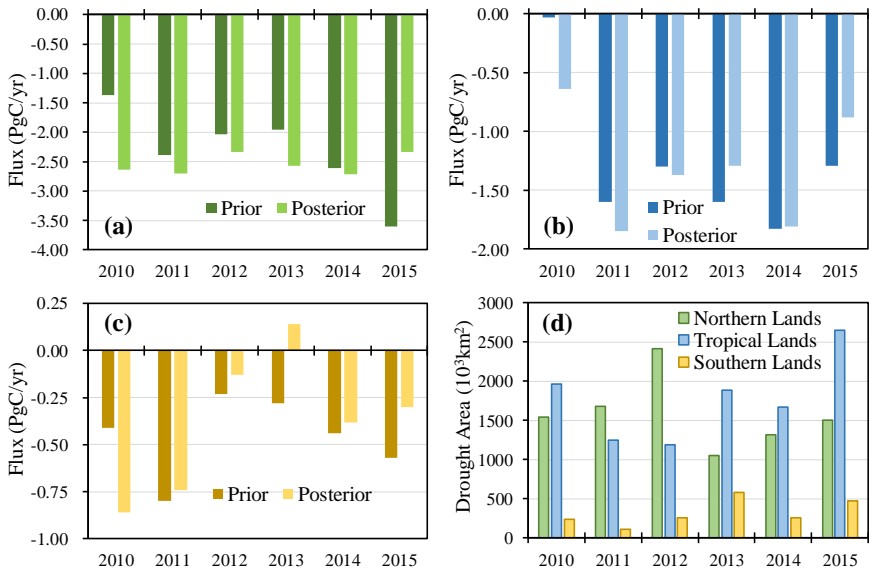

**Figure 9**. Prior and posterior interannual variations of the BIO fluxes in (a) Northern Lands, (b) Tropical Lands, and (c) Southern Lands, respectively, and (d) severe drought areas of above 3 regions.

Drought is one of the most important factors that affect terrestrial carbon sinks, and generally, severe drought will significantly reduce carbon sinks (e.g., Ma et al., 2012; Zhao and Running, 2010; Ciais et al., 2005; Gatti et al., 2014; Phillips et al., 2009; Vicente-Serrano et al., 2013). Previous studies (e.g., Liu et al., 2018) have used the GOSAT $XCO_2$ retrievals to infer the impact of droughts on terrestrial ecosystem carbon uptake anomalies. Figure 9d shows the severe drought areas (SDAs) in the 3 large regions every year, which were calculated according to the monthly Standardised Precipitation-Evapotranspiration Index at 12-month time scales (SPEI12) (Beguería et al., 2010). Here, the database of SPEIbase v2.5 is used, and the severe drought is defined as SPEI12 less than -1.5 (Paulo et al., 2012). In addition, only the severe drought that happens in forests, shrubs and crops are counted in this study. It could be found that the posterior fluxes have better correlations with the SDAs in all 3 regions, i.e. a larger SDA leads to a weaker carbon sink, and vice versa. The correlation coefficients between carbon sinks and SDAs in Northern Lands, Tropical Lands and





Southern Lands increase from prior values of -0.1, -0.25 and -0.44 to -0.53, -0.67 and -
0.76, respectively, indicating that the inversion has improved the interannual variations
of BIO fluxes in large scales. In addition, strong El Niño event happened during
2015~2016, and many researches have studied the responses of tropical land carbon
fluxes to this strong El Niño event (e.g., Wang et al., 2018; Liu et al., 2017; Bastos et
al., 2018; Koren et al., 2018). Liu et al. (2017) found that relative to the 2011 La Niña,
the pantropical biosphere released $2.5 \pm 0.34$ PgC more carbon into the atmosphere in
2015. Bastos et al. (2018) showed a smaller difference of carbon fluxes between 2015
and 2011 using both bottom-up and top-down approaches, which was in the range of
$-0.7 \sim -1.9$ PgC yr$^{-1}$. In this study, compared with the prior, our inversion significantly
enhances the difference between 2011 and 2015 (Figure 9b), and shows that 2015
released 1.35 PgC more than 2011 in the pantropical region (defined as Liu et al., 2017),
which is much smaller than Liu et al.'s result, but agree well with the result of Bastos
et al. (2018).
Moreover, Figure 10 shows the prior and posterior interannual variations of the
BIO fluxes on the 11 TRANSCOM land regions. In North America, including
Temperate North America and Boreal North America, the prior fluxes show an upward
trend, while the posterior fluxes show a downward trend. In Boreal Asia and Temperate
Asia, there are significant upward trends for the prior fluxes, but no significant trends
are found in the posterior fluxes. In Temperate South America, although the prior and
posterior fluxes show trends of weakening first and then increasing, the years in which
the carbon sink is weakest are not consistent: the prior flux is weakest in 2012, while
the posterior one is in 2013. Similarly, in northern Africa, the prior and posterior fluxes
show a trend of increasing and then decreasing, but the prior flux is the largest in 2014,
while the posterior one is strongest in 2011. In other regions, i.e., Tropical South
America, Tropical Asia, Southern Africa, Australia and Europe, the trends between the
prior and posterior fluxes are similar, especially in Tropical South America and Tropical
Asia, the prior and posterior fluxes are very close every year. Among them, in Southern
Africa and Australia, the posterior fluxes have more significant interannual variations



than the prior fluxes, and in Europe, the posterior sink is much weaker in 2015, and
stronger in 2010 and 2013 than the prior one.

The same as above, we also investigate the relationships between the interannual

variations of carbon sinks and SDAs in the 11 TRANSCOM land regions. As shown in
Table 4, in Temperate South America, Boreal Asia, and Europe, the posterior sinks have
a better correlation with the SDAs than the prior sinks, especially in Europe, the
correlation coefficient increases from a prior value of -0.33 to -0.85. However, in other
regions, there is no obvious improvement, and in some regions, the relationships are
even getting worse, such as Boreal North America, Temperate North America, Northern
Africa and Southern Africa. One possible reason is that there are usually higher annual
mean temperatures in drought years, which might extend the growing season of
vegetation, thereby enhance the carbon uptake and offset the impacts of drought. A
previous study (Wolf et al., 2016) showed that in 2012, Temperate North America
experienced an extreme summer drought event, and along with the warmest spring on
record. They quantified the impact of this climate anomaly on the carbon cycle and
concluded that the warm spring largely increased spring carbon uptake, and thus
compensated for reduced carbon uptake induced by the summer drought. Liu et al.
(2018) reported that because of the compensating effect of the carbon flux anomalies
between northern and southern US in 2011 and between spring and summer in 2012,
the annual carbon uptake decreased by $0.10\pm0.16$ PgC in 2011, and increased by
$0.10\pm0.16$ GtC in 2012 over US compared with the averaged state. In this study,
compared with the mean flux during 2010-2015, the carbon sink in Temperate North
America decreased by 0.09 PgC $yr^{-1}$ in 2011, and increased by 0.14 PgC $yr^{-1}$ in 2012,
which is very close to the result of Liu et al. (2018). In Australia, both the prior and
posterior fluxes have very good relationships with the SDAs. The significantly
enhanced carbon uptake during 2010-2012 is consistent with the finding in Detmers et
al. (2015), who inferred an even stronger carbon sink of $-0.77\pm0.10$ PgC $yr^{-1}$ from the
end of 2010 to early 2012 using the GOSAT $XCO_2$ product, and they confirmed that
this enhanced sink is related to the strong La Niña episode, which brought a record-



breaking amount of precipitation, resulting in an enhanced growth of vegetation. In
Tropical South America, the impacts of the 2010 drought on the carbon uptake over
Amazon have been extensively studied (e.g., Doughty et al., 2015; Gatti et al., 2014;
van der Laan-Luijkx et al., 2015). 2010 is a drought year, while 2011 is a wet year in
the Amazon region, compared to 2011, Gatti et al. (2014) estimated the no-fire carbon
exchange was reduced by 0.22 PgC yr$^{-1}$, van der Laan-Luijkx et al. (2015) derived a
decrease of biospheric uptake ranging from 0.08 to 0.26 PgC yr$^{-1}$, and Doughty et al.
(2015) concluded that drought suppressed Amazon-wide photosynthesis by 0.23–0.53
PgC yr$^{-1}$. In this study, our inversion reduces the difference of carbon uptake between
2010 and 2011 from a prior of 0.62 PgC yr$^{-1}$ to 0.28 PgC yr$^{-1}$, which is much more
consistent with the previous estimates.
Carbon uptake occurs mainly through photosynthesis of vegetation leaves. Leaf
area index (LAI) is a measure of leaf area per unit area. Buchmann and Schulze (1999)
shown that there are strong relationships between the interannual changes of carbon
uptake and LAI in grasslands, C4 crops, and coniferous forests, but no significant
relationship in broad-leaved forests; Chen et al. (2019) also showed that from 1981 to
2016, the increase in LAI contributed significantly to the increase in global BIO carbon
sinks. Therefore, we further investigate the relationships between the interannual
changes of carbon sinks and LAIs in the 11 TRANSCOM regions (Table 4). Here, the
LAI data are from the GIMMS LAI3g product, which has a spatial resolution of 1/12
degree and a time interval of 15 days (Zhu et al., 2013). As shown in Table 4, in Boreal
North America, Temperate North America, Northern Africa and Southern Africa,
compared with the prior fluxes, there are better relationships between the posterior
carbon sinks and LAIs, the correlation coefficients increase from prior values of -0.4,
0.31 and 0.35 to 0.62, 0.73 and 0.90 respectively, suggesting that the inversion of this
study may also improve the interannual variations of carbon sinks in these 4 regions at
a certain extent.





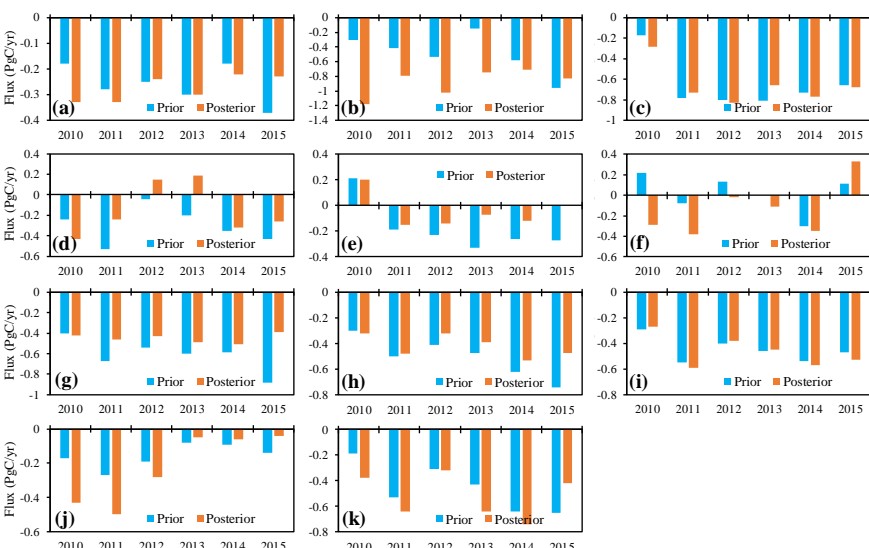

**Figure 10**. Prior and posterior interannual variations of the BIO fluxes on (a) Boreal

North America, (b) Temperate North America, (c) Tropical South America, (d)

Temperate South America, (e) Northern Africa, (f) Southern Africa, (g) Boreal Asia,

(h) Temperate Asia, (i) Tropical Asia, (j) Australia, and (k) Europe

**Table 4**. Correlation coefficients of severe drought areas (SDAs) and regional mean

LAI with the BIO sinks in each region

| Regions | SDA | | LAI | |
|---|---|---|---|---|
| | Prior | Posterior | Prior | Posterior |
| Boreal North America | -0.29 | 0.36 | -0.4 | 0.62 |
| Temperate North America | -0.54 | -0.27 | 0.31 | 0.73 |
| Tropical South America | -0.1 | -0.2 | 0.64 | 0.49 |
| Temperate South America | -0.41 | -0.74 | 0.72 | 0.24 |
| Northern Africa | 0.51 | 0.2 | 0.81 | 0.89 |
| Southern Africa | -0.53 | 0.41 | 0.35 | 0.9 |
| Boreal Asia | -0.17 | -0.35 | 0.49 | 0.1 |
| Temperate Asia | 0.33 | 0.33 | 0.55 | 0.38 |
| Tropical Asia | -0.03 | 0.16 | 0.69 | 0.71 |
| Australia | -0.85 | -0.73 | 0.88 | 0.83 |
| Europe | -0.33 | -0.85 | 0.85 | 0.58 |



## 5. Summary and Conclusions


In this study, we upgrade the GCAS system to GCASv2 with new assimilation
algorithms, procedures and a localization scheme, a higher assimilation parameter
resolution, and the ability to assimilate $XCO_2$ retrievals. Then, we use the GOSAT
$XCO_2$ retrievals to constrain terrestrial ecosystem and ocean carbon fluxes from May
1, 2009 to Dec 31, 2015, using the GCASv2 system. We compare the simulated prior
and posterior $XCO_2$ against the corresponding GOSAT $XCO_2$ retrievals to test the
effectiveness of the assimilation system and evaluate the posterior carbon fluxes by
comparing the posterior $CO_2$ mixing ratios against observations from 52 surface flask
sites. The distribution and interannual variations of the posterior carbon fluxes at both
global and regional scales from 2010 to 2015 are shown and discussed.
Compared with the GOSAT $XCO_2$ retrievals, the global mean BIAS and RMSE
decrease from prior values of 1.8±1.3 and 2.2 ppm to -0.0±1.1 and 1.1 ppm, respectively,
indicating that the GCASv2 system works well with the GOSAT $XCO_2$ retrievals.
Independent evaluations using surface flask $CO_2$ concentrations showed that the
posterior carbon fluxes could significantly improve the modeling of atmospheric $CO_2$
concentrations, with the global mean BIAS and RMSE decreasing from prior values of
1.6±1.8 and 2.4 ppm to -0.5±1.8 and 1.9 ppm, respectively. The large negative biases
are mainly distributed in North America, Europe, indicating the overestimates of carbon
sinks over these areas. Evaluations also show that the biases gradually increase along
with the time in most tropical and southern hemisphere ocean sites, but no accumulation
is found at most land sites, indicating that globally, the carbon sinks may be
overestimated every year, but in different lands, the deviations of the estimates may
differ each year.
Globally, the mean annual BIO carbon sink and the interannual variations
inferred in this study are very close to the estimates of CT2017 during the study period,
and the estimated mean AGR and interannual changes are also very close to the
observations, with mean annual bias of -0.11 PgC yr$^{-1}$. Regionally, the inversion shows



that in the northern lands, the carbon sink of Temperate North America is the strongest,
and those in Boreal Asia, Temperate Asia and Europe are also very strong and
comparable; in the tropics, there are strong sinks in Tropical South America and
Tropical Asia, but a very weak sink in Africa. These distributions are significantly
different from the estimates of CT2017, probably due to the different prior fluxes and
$CO_2$ observations used for inversion. However, our estimates in most regions or
continents are comparable or in the range of previous bottom-up estimates. The
inversion also changed the interannual variations of carbon sinks in most TRANSCOM
and hemisphere scale land regions, leading to their better relationship with the
variations of severe drought or LAI, indicating that the inversion with GOSAT $XCO_2$
retrievals may help to better understand the interannual variations of regional carbon
fluxes.

## Data availability

The code of GCASv2 system and the inversion results of this study are available to the
community and can be accessed upon request from Fei Jiang (jiangf@nju.edu.cn) at
Nanjing University.

## Author contributions

FJ, JC and WJ designed the research; FJ run the model, analyzed the results and wrote
the paper; HW handled the GOSAT $XCO_2$ retrievals; WH analyzed the drought data;
XL run the BEPS model; FJ lead the update of the GCAS system, and XT, HW, JW, SF,
GL, ZC, SZ, JL, WH, and MW participated in it; RL, PS and PK provided the surface
$CO_2$ observations; JC, WJ and HW participated in the discussion of the inversion results
and provided input on the paper for revision before submission.

## Competing interests

The authors declare that they have no conflict of interest.





## Acknowledgements

This work is supported by the National Key R&D Program of China (Grant No: 2016YFA0600204). We acknowledge all atmospheric data providers to obspack_co2_1_GLOBALVIEWplus_v5.0_2019_08_12. We especially thank Pieter Tans, Ed Dlugokencky, Kenneth Schuldt at NOAA ESRL, USA and Ray Langenfelds, Paul Steele, Paul Krummel at CSIRO, Australia for their great efforts on $CO_2$ observations and data distributions. CarbonTracker CT2017 results are provided by NOAA ESRL, Boulder, Colorado, USA, from the website at http://carbontracker.noaa.gov. The GOSAT data are produced by the OCO project at the Jet Propulsion Laboratory, California Institute of Technology, and obtained from the data archive at the NASA Goddard Earth Science Data and Information Services Center. We are also grateful to the High-Performance Computing Center (HPCC) of Nanjing University for doing the numerical calculations in this paper on its blade cluster system.

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
