# Peer review of "Regional CO2 Fluxes during 2010-2015 Inferred from GOSAT XCO2"

_Atmospheric Chemistry and Physics, 2020_

## Referee Comment (RC1) · Anonymous Referee #2 · 15 Sep 2020

General comments: In this study, Jiang et al. upgraded the Global Carbon Assimilation System (GCAS) with new assimilation algorithms, a localization scheme, and a higher assimilation parameter resolution, namely GCASv2. The global terrestrial ecosystem (BIO) and ocean (OCN) carbon fluxes from 2009 to 2015 were constrained by the GOSAT ACOS XCO2 retrievals. Following this, the posterior carbon fluxes from 2010 to 2015 were evaluated using 52 surface flask observations. The errors in the posterior carbon fluxes in the new inversion system were compared to those in a previous version. The authors indicated that the pattern of regional carbon sinks was significantly different from previous studies (CT2017). The inter-annual variations of carbon fluxes in most land regions, and the relationship with the changes of severe drought area the

plant indexes, and drought were re-visited. These results are interesting. However, the improvement of the inversion methodology is not presented, and the reduction of the uncertainty by the inversions remains unclear (Figure 3) in the current paper. I, therefore, recommend that this work cannot be published before the following comments are addressed.

Specific comments: Figure 3: What is the source for error bars in these two plots? Are they coming from the uncertainty in the prior and posterior estimates? If yes, it seems that the uncertainty is not reduced from the prior estimates to the posterior estimates. One main purpose of inversion is to reduce the uncertainty in the prior estimates. If the uncertainty is not reduced, the effectiveness of the inversions should be evaluated.

Figures 4/5: Evaluation of the reduction of the uncertainty from the prior estimates to the posterior estimates is more important than evaluation of the bias itself for an inversion system.

Tables 2/3: What is the uncertainty for the prior and posterior estimates?

Line 473-488: What is the uncertainty for the estimates from this study? To evaluate the effectiveness of an inversion system, the uncertainty of the posterior estimates is more important than the central value. Such information is missing in the current manuscript, which is better considered / discussed in previous studies (e.g. the literature cited in line 586).

Figures 7/9/10: What is the uncertainty for the prior and posterior estimates?

Figure 1: The authors suggested that a new assimilation scheme is developed in this paper. Why not directly compare the flow charts between the GCAS and GCASv2 systems and show the difference?

Line 124: It seems that a major advance of GCASv2 against GCAS is that "In the second step, the MOZART-4 model is run again using the optimized fluxes of Xa, to generate new $CO_2$ concentrations for the initial field of the next DA window. This DA

flow chart is different from the previous version of GCAS, which runs the MOZART-4 model only once, and optimizes the fluxes and the initial field of the next window synchronously." However, I do not understand how this improves the inversion system. The old GCAS system produces the posterior global gridded carbon fluxes, which were used as prior fluxes as input to any other forward models to simulate the CO2 field. If the difference of GCASv2 was just that the posterior global gridded carbon fluxes were used by MOZART-4 to simulate the CO2 field, I cannot see how and why the inversing methodology is improved.

Line 143: It seems that the carbon emission from cement production, a large part of CO2 source, is missed in this inversion system. This could be a big weakness of the current system.

Line 143: What is the relationship between BIO and FIRE? Biomass sequestrates carbon from the atmosphere, and releases CO2 in biomass burning. Should FIRE be a part of BIO?

Line 147: "FOSSIL and FIRE fluxes are assumed to have no errors, only BIO and OCN fluxes are optimized in an assimilation system". I do not think that this is the case in other inversion systems: (1) It needs clear justification by summarizing and tabulating the methodology in the literature. (2) The difference relative to a system with errors considered for FOSSIL and FIRE need to be calculated to show how much the conclusion of the present study are sensitive to this assumption.

Line 209: How does GCASv2 consider the spatial representativeness errors in the inversion system?

Line 238: How many sites are subject to this spurious noise? Are these sites excluded from the inversion system? How much does removing data at these sites influence the inversion fluxes?

Technical corrections:

[Figure]

Line 38: "BIAS" is not defined before it is used.

Line 63: "However, their carbon uptakes have significant spatial differences and inter-annual variations." References are needed.

Line 95: "However, so far, on the one hand, most studies focused on the impact of GOAST XCO 2 retrievals on the inversion of surface carbon fluxes, but in many regions, there are still large divergences for carbon sinks between different inversions with the same GOSAT data or between inversions with GOSAT and in situ observations (Chevallier et al., 2014)". Is only one study considered and cited?

Line 102. References are needed.

Line 255: The references for the two emission inventories of FOSSIL and FIRE are out of date. ODIAC and GFEDv4 have been updated recently.

Line 270: "The BIO carbon flux, which is the most important prior carbon flux". Why is the prior carbon flux of BIO more important than FOSSIL and FIRE to an inversion system?

Line 340: When the averages of the modeled and the observational values/retrievals are equal, BIAS is zero, even if all data are distant to the 1:1 line in the comparison. BIAS cannot effectively evaluate the performance of the model by showing how much the modeled values/retrievals agree with the observational values/retrievals. The average of absolute difference between the modeled and the observational values/retrievals is needed.

Line 360: Does the study of Wang et al. (2019) account for the uncertainty in FOSSIL and FIRE?

Line 448: What is "impact of accumulation"?

Figures 3/4: "Biases" in the caption is easily confused with "BIAS" defined in equation 10.

Table 1: BIAS cannot evaluate the performance of the model by showing how much the modeled values/retrievals agree with the observed values/retrievals.

[Figure]

---

## Referee Comment (RC2) · Anonymous Referee #1 · 23 Sep 2020

General comments.

Authors present estimates of regional carbon dioxide flux variability based on assimilating GOSAT satellite observations of CO2 with ensemble-based data assimilation system. The estimated CO2 fluxes where evaluated by comparison to indexes of climate variability, and published top-down and bottom-up estimates. The analysis of the carbon cycle variability and comparison with data on climate variability makes a strong point of the study. On the other hand, the description of the ensemble-based data assimilation system can be improved. The paper is well written and can be accepted after minor revisions addressing the review suggestions.

[Figure]

Detailed comments.

Lines 130-139 Suggest clarifying, what becomes a state vector to optimize, currently it is implicit. Some details emerge much later on Lines 358-366, when uncertainties are discussed.

Lines 232-236 The logic behind selecting 1-week data assimilation window doesn't look solid, as the other ensemble-based assimilation systems use longer window in order of 12 weeks, ( Peters et al. 2005, Feng et al. 2009, Jacobson et al. 2020. The notice that there was a problem reproducing $CO_2$ growth rate with a longer window in Zhang et al (2015) doesn't look like a strong argument, if considered in comparison with other studies.

Technical corrections

Lines 119-120 Need to clarify, written that fluxes "are perturbed with a Gaussian random distribution" – better add more detail on whether perturbation is applied independently to each grid or over regions.

Line 216 As resolutions of the transport model and fluxes are apparently different, suggest writing which of them are referred as 'model grids'.

Line 584 Revise 'a very stronger carbon sink' as 'a stronger carbon sink' or 'a very strong carbon sink'

Line 594 Suggest revising 'weak' to 'weaker'

References

Feng, L., Palmer, P. I., Bösch, H., and Dance, S.: Estimating surface $CO_2$ fluxes from space-borne $CO_2$ dry air mole fraction observations using an ensemble Kalman Filter, Atmos. Chem. Phys., 9, 2619–2633, https://doi.org/10.5194/acp-9-2619-2009, 2009.

Peters, W., Miller, J. B., Whitaker, J., Denning, A. S., Hirsch, A., Krol, M. C., Zupanski, D., Bruhwiler, L., and Tans, P. P. (2005), An ensemble data assimilation system to

estimate CO2 surface fluxes from atmospheric trace gas observations, J. Geophys. Res., 110, D24304, doi:10.1029/2005JD006157.

---

## Author Comment (AC1) · 30 Nov 2020

**Referee #1**

We would like to thank the anonymous referee for his/her comprehensive review and valuable suggestions. These suggestions help us to present our results more clearly. In response, we have made changes according to the referee's suggestions and replied to all comments point by point. All the page and line number for corrections are referred to the revised manuscript, while the page and line number from original reviews are kept intact.

General comments.

Authors present estimates of regional carbon dioxide flux variability based on assimilating GOSAT satellite observations of $CO_2$ with ensemble-based data assimilation system. The estimated $CO_2$ fluxes where evaluated by comparison to indexes of climate variability, and published top-down and bottom-up estimates. The analysis of the carbon cycle variability and comparison with data on climate variability makes a strong point of the study. On the other hand, the description of the ensemble-based data assimilation system can be improved. The paper is well written and can be accepted after minor revisions addressing the review suggestions.

Detailed comments.

Lines 130-139 Suggest clarifying, what becomes a state vector to optimize, currently it is implicit. Some details emerge much later on Lines 358-366, when uncertainties are discussed.

Response: Thank you for this suggestion. In this study, the terrestrial ecosystem (BIO) and ocean (OCN) carbon fluxes are treated as state vector and optimized. Indeed, as you said, the state variables had been mentioned in two places in the article. The first place is in the section of system description, and the second is in the section of "Experimental Design". In the first place, we are introducing the current system (GCASv2) that we have improved, we set 4 state vector schemes in this system for different applications: 1) only the BIO flux is state vector; 2) both BIO and OCN fluxes are treated as state vectors; 3) the BIO, OCN and FOSSIL fluxes are optimized at the same time; and 4) only net flux is optimized. In this study, we chose to optimize both BIO and OCN, which were introduced in the section of "Experimental Design". To further clarify the state vector of this study, we added a sentence of "*In this study, the second scheme was selected.*" at the end of the 2nd paragraph in section 2.1 (see Line 178, Page 7).

Lines 232-236 The logic behind selecting 1-week data assimilation window doesn't look solid, as the other ensemble-based assimilation systems use longer window in order of 12 weeks, (Peters et al. 2005, Feng et al. 2009, Jacobson et al. 2020. The notice that there was a problem reproducing $CO_2$ growth rate with a longer window in Zhang et al (2015) doesn't look like a strong argument, if considered in comparison with other studies.

Response: Many thanks for this suggestion. We have added more discussions about the assimilation window, and shown the mean observation (only GOSAT $XCO_2$) number (Figure S2) during the study period that each grid could have within the 1 week assimilation window and the 3000 km localization scale. We also conduct a test in the year of 2010 for different DA windows (1, 2 and 4 weeks) and evaluate the posterior results using surface observations

(see Table 1). We have revised that paragraph (see Lines 303-307, Lines 309-340, Pages 11-12) as follows:

"The DA window is set to one week in GCASv2, which is the same as before. Theoretically, a longer DA window is better, because $CO_2$ is a stable species. The longer window, the farther $CO_2$ will be transported. In this way, more observation stations will sense the flux change of one area, and thus more observations can be used to optimize the flux of that place. Therefore, many previous ensemble-based assimilation systems used a longer DA window (e.g., Peters et al. 2005, Feng et al. 2009, Jacobson et al. 2020). However, the farther away, the weaker signal the stations can sense. Bruhwiler et al. (2005) clearly shown that a pulse traveling from a faraway place would contribute relatively little signal compared to recent pulses from nearby source regions. In addition, Limited by the method of EnKF, this weak signal will be masked by the method's own unphysical signal (spurious correlation), and in order to reduce this influence, we must increase the ensembles, thereby greatly increasing the computational cost. Miyazaki et al. (2011) tested the differences of 3 days and 7 days DA windows, and pointed that with a longer DA window, more observation data will be available to constrain the surface flux, but a longer window can make the effect of model error more obvious. Thus, the assimilation result can be improved as long as the observations with spurious correlations can be neglected. However, spurious correlations can be more serious with increases in the DA window, because of a limited number of ensembles. As a result, a longer window is not necessarily better than a shorter window system. To avoid the influence of spurious signals, Kang et al. (2012) used a very short DA window (6 hours) in their assimilation system (LETKF_C) and pointed out that the flux inversion with a long window (3 weeks) is not as accurate as the one obtained with a 6 h DA window, particularly in smaller-scale structures. During the development of GCASv1, Zhang et al. (2015) tested different DA windows and found that the longer the window, the larger optimized terrestrial carbon sink will be, resulting in a smaller optimized annual atmospheric $CO_2$ growth rate as compared to the observed rate. Considering the fact that at present, due to the release of satellite $XCO_2$ retrievals like GOSAT and OCO-2, the atmospheric $CO_2$ observations and coverages have increased significantly compared to before, which means that we do not need to extend the DA window to include more observation data now. Figure S2 shows the mean super observation (see section 2.1.1, only GOSAT $XCO_2$) numbers during the study period that each grid could have within the 1-week DA window and a localization scale (3000 km, see the next paragraph). In most land areas and pan-tropical waters, each grid can already have more than 3 super observations. On average, each grid over the land could has 4 super observations. Two sensitivity tests in 2010 were conducted using 2- and 4- weeks DA windows but the same localization scale, the results are shown in Table S3. When the length of DA window increases from 1 week to 4 weeks, the mean super observation number increases from 4 to 9, accordingly, the inverted global BIO flux increased from -4.16 PgC yr$^{-1}$ to -4.49 PgC yr$^{-1}$, resulting in a larger deviation of the simulated and observed atmospheric $CO_2$ growth rate (AGR) and larger simulation error against the surface observations. Therefore, we still use the 1-week DA window in GCASv2."

[Figure]

**Figure 1**. Mean observation numbers within a DA window (1 week) during May 2009 ~ Dec 2015 (This figure has been added in the revised Supporting Information, and named as Figure S2)

**Table 1**. Results of sensitivity tests in the year of 2010 (1week, 2weeks and 4weeks are three additional experiments using 1 week, 2 weeks, and 4 weeks assimilation windows, respectively) (This Table has been added in the revised Supporting Information, and named as Table S4)

| | | Prior | 1 week | 2 weeks | 4 weeks |
|---|---|---|---|---|---|
| Super Obs. Num. per window | Total | - | 730 | 1039 | 1360 |
| | Each grid over land | - | 4 | 6 | 9 |
| Global Flux (PgC/yr) | BIO | -2.07 | -4.16 | -4.46 | -4.49 |
| | OCN | -2.08 | -2.33 | -2.32 | -2.35 |
| | FOSSIL | 9.07 | 9.07 | 9.07 | 9.07 |
| | Net | 7.25 | 4.91 | 4.62 | 4.55 |
| Regional Flux (PgC/yr) | North America Boreal | -0.29 | -0.43 | -0.41 | -0.35 |
| | North America Temperate | -0.42 | -1.25 | -1.75 | -2.41 |
| | Tropical South America | -0.17 | -0.26 | -0.32 | -0.27 |
| | Temperate South America | -0.24 | -0.4 | -0.36 | -0.19 |
| | Northern Afirca | 0.21 | 0.32 | 0.36 | 0.62 |
| | Southern Africa | 0.22 | -0.3 | -0.59 | -1.04 |
| | Boreal Asia | -0.4 | -0.46 | -0.3 | 0.11 |
| | Temperate Asia | -0.3 | -0.29 | -0.15 | -0.06 |

| | | | | | |
|---|---|---|---|---|---|
| | Southeast Asia | -0.29 | -0.23 | -0.21 | -0.2 |
| | Australia | -0.17 | -0.4 | -0.48 | -0.53 |
| | Europe | -0.19 | -0.41 | -0.21 | -0.12 |
| independent evaluation | BIAS | 1.43 | -0.44 | -0.4 | -0.38 |
| | MAE | 1.92 | 1.37 | 1.39 | 1.51 |
| | RMSE | 2.36 | 2.11 | 2.18 | 2.39 |
| Deviation from the observed AGR (PgC yr$^{-1}$) | | 2.08 | -0.26 | -0.55 | -0.62 |

Technical corrections

Lines 119-120 Need to clarify, written that fluxes "are perturbed with a Gaussian random distribution" – better add more detail on whether perturbation is applied independently to each grid or over regions.

Response: Thank you! We have rewritten that sentence (see Lines 123-124, Page 5), as follows:

"the prior fluxes of $X^b$ in each grid are independently perturbed with a Gaussian random distribution"

Line 216 As resolutions of the transport model and fluxes are apparently different, suggest writing which of them are referred as 'model grids'.

Response: Thanks for this suggestion. We have changed 'model grids' as 'transport model grids' (see Lines 268 and 269, Page 10).

Line 584 Revise 'a very stronger carbon sink' as 'a stronger carbon sink' or 'a very strong carbon sink'

Response: Thanks! We have changed 'a very stronger carbon sink' as 'a very strong carbon sink' (see Line 857, Page 29).

Line 594 Suggest revising 'weak' to 'weaker'

Response: Thanks! We have changed 'weak' to 'weaker' (see Line 870, Page 30).

---

## Author Comment (AC2) · 30 Nov 2020

**Referee #2**

We thank the anonymous referee for his/her valuable comments and constructive suggestions. We have made changes according to the referee's suggestions and replied to all comments point by point. All the page and line number for corrections are referred to the revised manuscript, while the page and line number from original reviews are kept intact. The references related to the responses are listed in the end of this document.

General comments:

In this study, Jiang et al. upgraded the Global Carbon Assimilation System (GCAS) with new assimilation algorithms, a localization scheme, and a higher assimilation parameter resolution, namely GCASv2. The global terrestrial ecosystem (BIO) and ocean (OCN) carbon fluxes from 2009 to 2015 were constrained by the GOSAT ACOS XCO2 retrievals. Following this, the posterior carbon fluxes from 2010 to 2015 were evaluated using 52 surface flask observations. The errors in the posterior carbon fluxes in the new inversion system were compared to those in a previous version. The authors indicated that the pattern of regional carbon sinks was significantly different from previous studies (CT2017). The inter-annual variations of carbon fluxes in most land regions, and the relationship with the changes of severe drought area the plant indexes, and drought were re-visited. These results are interesting. However, the improvement of the inversion methodology is not presented, and the reduction of the uncertainty by the inversions remains unclear (Figure 3) in the current paper. I, therefore, recommend that this work cannot be published before the following comments are addressed.

Specific comments:

Figure 3: What is the source for error bars in these two plots? Are they coming from the uncertainty in the prior and posterior estimates? If yes, it seems that the uncertainty is not reduced from the prior estimates to the posterior estimates. One main purpose of inversion is to reduce the uncertainty in the prior estimates. If the uncertainty is not reduced, the effectiveness of the inversions should be evaluated.

Response: Thank you for this suggestion. The error bars represent the standard deviations of all biases at each latitude and each site, respectively. Indeed, the uncertainty reduction is very important for an inversion study. We analyzed the uncertainty reduction rate (UR), and added a section of "4.2 Uncertainty reduction" in the revised manuscript (see Lines 663 – 705, Pages 21 - 23). The annual mean URs of the BIO fluxes over different TRANSCOM regions are in the range of 6% ~ 27%, with global mean of 17%. The highest monthly UR is 51% in temperate South America.

Figures 4/5: Evaluation of the reduction of the uncertainty from the prior estimates to the posterior estimates is more important than evaluation of the bias itself for an inversion system.

Response: Thank you for this suggestion. We have analyzed the reductions of the uncertainties from the prior estimates to the posterior estimates and added a section of "4.2 Uncertainty reduction" in the revised manuscript (see Lines 663 – 705, Pages 21 - 23).

Tables 2/3: What is the uncertainty for the prior and posterior estimates?
Response: Thank you! We have added the uncertainties of the prior and posterior estimates in the revised manuscript (see Lines 719 – 723, Page 24 and Line 828, Page 28).

Line 473-488: What is the uncertainty for the estimates from this study? To evaluate the effectiveness of an inversion system, the uncertainty of the posterior estimates is more important than the central value. Such information is missing in the current manuscript, which is better considered / discussed in previous studies (e.g. the literature cited in line 586).
Response: Thank you for this suggestion! As shown above, we have analyzed the uncertainty reductions and added a section of "4.2 Uncertainty reduction" in the revised manuscript (see Lines 663 – 705, Pages 21 - 23).

Figures 7/9/10: What is the uncertainty for the prior and posterior estimates?
Response: Thank you for this suggestion. We have added the prior and posterior uncertainties in Figures 7, 9 and 10, which are named as Figure 8, 10 and 11 in the revised manuscript (see Lines 893-896, Page 31; Lines 934-937, Page 33; and Lines 1045-1049, Page 37).

Figure 1: The authors suggested that a new assimilation scheme is developed in this paper. Why not directly compare the flow charts between the GCAS and GCASv2 systems and show the difference?
Response: Many thanks for this suggestion. We have modified Figure 1 and given the differences in the flow charts between GCASv1 and GCASv2 (see Lines 147-148, Page 6).

Line 124: It seems that a major advance of GCASv2 against GCAS is that "In the second step, the MOZART-4 model is run again using the optimized fluxes of Xa, to generate new CO2 concentrations for the initial field of the next DA window. This DA flow chart is different from the previous version of GCAS, which runs the MOZART-4 model only once, and optimizes the fluxes and the initial field of the next window synchronously." However, I do not understand how this improves the inversion system. The old GCAS system produces the posterior global gridded carbon fluxes, which were used as prior fluxes as input to any other forward models to simulate the CO2 field. If the difference of GCASv2 was just that the posterior global gridded carbon fluxes were used by MOZART-4 to simulate the CO2 field, I cannot see how and why the inversing methodology is improved.
Response: Thank you for this comment. Indeed, as you said, the descriptions of the differences between GCASv2 and GCASv1 are rather vague. We have revised Section 2.1 to further clarify their differences. The main differences between GCASv2 and GCASv1 are as follows:

1) Optimization of the initial field of each window. In GCASv1, it is directly optimized using the observations, while in GCASv2, it is simulated using the posterior fluxes of the previous window. The advantage of this method in GCASv2 is that the assimilation errors could be

transported from one window to the next. If the fluxes are overestimated in one window because of some reasons, by this method, they will affect the concentrations of the next window, thereby the posterior fluxes of the next window will compensate the overestimations. While in GCASv1, since the initial field of each window is directly optimized using the observations, which means in each window, there are relatively perfect initial fields, the inversions of each window are independent, and the amount of overestimation or underestimation in one window will continue to accumulate until the end, leading to an overall overestimation or underestimation. In addition, due to the perfect initial field, the differences between the simulated and observed concentrations are only contributed by the errors in the prior fluxes of current window, resulting in a relatively smaller model – data mismatch, so as to weaken the assimilation benefits on fluxes. This difference is given in Lines 128 – 143, Page 5 in the revised manuscript.

2) State vector. In GCASv1, only BIO is state vector, while in GCASv2, we set 4 state vector schemes for different applications: 1) only the BIO flux is state vector; 2) both BIO and OCN fluxes are treated as state vectors; 3) the BIO, OCN and FOSSIL fluxes are optimized at the same time; and 4) only net flux is optimized. This difference is given in Lines 172 – 178, Page 7 in the revised manuscript.

3) Resolution of the state vectors. In GCASv1, the scaling factor $\lambda$ is defined in different land and ocean areas based on 22 TRANSCOM regions (Gurney et al., 2002) and 19 Olson ecosystem types, as in CarbonTracker (Peters et al., 2007), while in GCASv2, we change to use a $\lambda$ in each grid, meaning that for each grid, the perturbations of prior fluxes are independent, and the grid cell of $\lambda$ could be set freely. This difference is given in Lines 154-161, Page 6 in the revised manuscript.

4) observation data. In GCASv1, only flask/in situ observations were assimilated, while in GCASv2, we added a module to assimilate the satellite $XCO_2$ retrievals, and allow users to simultaneously or separately assimilate the flask/in situ concentrations and the $XCO_2$ retrievals. See Lines 186 – 201, Pages 7 -8 in the revised manuscript. Besides, a 'super-observation' approach is also adopted in GCASv2, See Lines 202-215, Page 8 in the revised manuscript.

5) assimilation algorithm, in GCASv2, we added another EnKF algorithm, i.e., EnSRF. See Lines 223-227, Page 9.

Line 143: It seems that the carbon emission from cement production, a large part of CO2 source, is missed in this inversion system. This could be a big weakness of the current system. Response: Sorry, that description is not accurate enough. The carbon emission from cement production has been included in this study. The fossil fuel carbon emissions are obtained from NOAA's CarbonTracker, version CT2017, which is an average of the Carbon Dioxide Information Analysis Center (CDIAC) product (Andres et al., 2011) and the Open-source Data Inventory of Anthropogenic CO2 (ODIAC) emission product (Oda et al., 2018). We have checked the document of CT2017 and the introduction of CDIAC database, compared the annual global fossil fuel emissions in our system with the global emissions from the CDIAC website (https://cdiac.ess-dive.lbl.gov/), and confirmed that the carbon emission from

cement production has been included in this study. We have changed the sentence of "…
atmosphere and ocean (OCN) carbon exchange, fossil fuel (FOSSIL) carbon emission and
biomass burning (FIRE) carbon emission…" to "*… atmosphere and ocean (OCN) carbon
exchange, fossil fuel **and cement production** (FOSSIL) carbon emission and biomass burning
(FIRE) carbon emission…*" (see Lines 166-167, Page 7)

Line 143: What is the relationship between BIO and FIRE? Biomass sequestrates carbon from
the atmosphere, and releases CO2 in biomass burning. Should FIRE be a part of BIO?
Response: Yes, biomass burning carbon emission is a part of terrestrial ecosystem carbon
flux. Terrestrial ecosystems uptake carbon through photosynthesis (GPP) and release carbon
through respiration (ER) and biomass combustion (FIRE). The BIO flux defined in this study
is the net flux of GPP and ER (ER-GPP). In many previous inversion studies, it is directly
defined as net ecosystem exchange [NEE = ecosystem respiration (ER) − gross primary
production (GPP)] (e.g., Hu et al., 2019; Peters et al., 2007, 2010), and the sum of NEE and
FIRE is defined as net biosphere exchange (NBE, Liu et al., 2017). In the revised manuscript,
we have changed the sentence of "… name terrestrial ecosystem (BIO) carbon flux, …" to
"*namely terrestrial ecosystem (BIO) carbon flux (i.e., net ecosystem exchange (NEE) =
ecosystem respiration (ER) − gross primary production (GPP)), …*" (see Lines 163-166,
Pages 6-7)

Line 147: "FOSSIL and FIRE fluxes are assumed to have no errors, only BIO and OCN
fluxes are optimized in an assimilation system". I do not think that this is the case in other
inversion systems: (1) It needs clear justification by summarizing and tabulating the
methodology in the literature. (2) The difference relative to a system with errors considered
for FOSSIL and FIRE need to be calculated to show how much the conclusion of the present
study are sensitive to this assumption.
Response: Thank you for this comment. Yes, there are considerable uncertainties for the fossil
fuel and biomass burning carbon emissions, which are about 6% and 20% for global mean,
respectively. Ideally, we would like the inversion to partition the deviations from the a-priori
fluxes among all the four type of carbon fluxes. NEE and ocean fluxes can, since they are
geographically separated, readily be accounted for in statistically independent deviation
terms. However, the inversion cannot be expected to distinguish between land biosphere
fluxes and fossil fuel emissions, because both are inextricably localized on land, and the $CO_2$
data alone do not discern fossil and non-fossil carbon (Rödenbeck et al., 2003). Therefore,
most inversion studies for surface carbon fluxes focused on the NEE and ocean fluxes, and
the fossil fuel and biomass burning were prescribed (e.g., Gurney et al., 2002, 2003; Peters et
al., 2007; Nassar et al., 2011; Feng et al., 2009; Monteil et al., 2020). As shown in Table 1, we
have reviewed a lot of studies, in which only Deng et al. (2014, 2015) considered the
uncertainties of fossil fuel and biomass burning carbon emissions, Liu et al. (2019) and Kang
et al. (2012) directly optimized the net carbon flux, and Some studies (Monteil et al., 2020,
Scholze et al., 2019) only optimized the NEE. Although Deng et al. (2014)'s state vector
includes emissions of $CO_2$ from fossil fuel combustion, when they reported their posteriori
flux estimates, they removed the a priori fossil fuel estimate from the reported total land flux.

As shown in section 2.1, we have added a scheme to simultaneously the fossil fuel and cement production carbon emissions in GCASv2. We have tried to use it to optimize the fossil fuel emissions in China. We tested different emission inventories, but GCASv2 did not make them converge, but only made the emissions of each inventory slightly lower. Therefore, we think that under the current resolution of atmospheric transport model, spatial coverage of observational data, and the assimilation settings, GCASv2 cannot optimize it well.

According to your suggestion, we added a sensitivity test for optimizing fossil fuel carbon emissions, using the same localization scheme as BIO and OCN, giving fossil fuels a global uncertainty of 5%. The results showed that the impact on both the inverted global and regional scale BIO fluxes are very small (Table 2).

The following sentences has been added in the revised manuscript:

"… and the FOSSIL and FIRE carbon emissions are kept intact *(the impact of this assumption on both the inverted global and regional BIO fluxes are very small (Table S4))*. Following Wang et al. (2019), …" (see Lines 558-560, Pages 16-17)

**Table 1**. a summary of the inversion methodology in the literature.

| System Name | Transport model/Res. | Assimilation method | Obs. | State Vector* | Reference |
|---|---|---|---|---|---|
| CT/CTE/CT-China | TM5,global 3x2, region, 1x1 | EnSRF | obspack | NEE, OCN | Peters et al., 2007; Peters et al., 2010; Zhang et al., 2014 |
| UoE | GEOS-Chem,4x5 | EnKF | in situ or GOSAT | NEE, OCN | Feng et al., 2009, 2016, 2017 |
| CAMS CO2 inversion system | LMDz,3.75x1.875 | variational | surface observations, GOSAT, OCO-2 | NEE, OCN | Chevallier, et al., 2019 |
| CCDAS | TM3,4x5 | 4D-Var | in situ CO2, SM, and L-VOD | NEE | Scholze et al., 2019 |
| Jena CarboScope | TM3,4x5 | time-independent Bayesian inversion | surface observations | NBE, OCN | Rödenbeck, 2005; Rödenbeck et al., 2003 |
| TransCom 3 inversions | 16 Atmospheric Transport Models,2.0x2.5 to 7.5x7.5 | Bayesian synthesis inversion | GLOBALVIEW data | NEE, OCN | Baker et al., 2006; Gurney et al., 2002, 2003 |

| | | | | | |
|---|---|---|---|---|---|
| Nasser et al., 2011 | GEOS-Chem,2x2.5 | time-independent Bayesian inversion | TES and surface flask measurements | NEE, OCN | Nassar et al., 2011 |
| EUROCOM (include 6 systems) | CHIMERE, FLEXPART, STILT, TM5, NAME/0.5x0.5 ~1x1 | Variational, EnKF, MCMC | flask | NEE OCN (4 prescribed) | Monteil et al., 2020 |
| Deng et al., 2007 | NIES,2.5x2.5 | Time-dependent Bayesian synthesis | GLOBALVIEW data | NEE, OCN | Deng et al., 2007 |
| Niwa et al., 2012 | NICAM-TM,~240 km | Time-dependent Bayesian synthesis | GLOBALVIEW, CONTRAIL | NEE, OCN | Niwa et al., 2012 |
| Miyazaki et al., 2011 | AGCM,2.8x2.8 | LETKF | OSSEs (GOSAT, CONTRAIL, and surface sites) | NEE, OCN | Miyazaki et al., 2011 |
| TM5-4DVAR inversion system | TM5,6x4 | 4D-Var | GOSAT | NEE, OCN | Basu et al., 2013 |
| GEOS-Chem-4DVAR inversion system | GEOS-Chem,4x5 | 4D-Var | GOSAT, Flask | NEE, OCN, FOSSIL, FIRE | Deng et al., 2014; 2016 |
| CMS-Flux inversion framework | GEOS-Chem,4x5 | 4D-Var | GOSAT, OCO-2, SIF | NBE, OCN | Liu et al., 2017 |
| LETKF_C | GEOS-Chem,4x5 | LETKF | OSSEs (GOSAT, CONTRAIL, and surface sites) | Net flux | Liu et al., 2019; Kang et al., 2012 |

*NEE: net ecosystem exchange, ecosystem respiration (ER) − gross primary production (GPP); NBE: net biosphere exchange, NEE + biomass burning carbon emission (FIRE); OCN: atmosphere - ocean carbon exchange; FOSSIL: fossil fuel and cement production carbon emission; Net flux: NEE + OCN + FOSSIL+ FIRE

**Table 2**. Results of sensitivity tests in the year of 2010 (Wfossil is an experiment with the

FOSSIL carbon emissions being synchronously optimized) (This Table has been added in the revised Supporting Information)

| | | Prior | 1 week | Wfossil |
|---|---|---|---|---|
| Super Obs. Num. per window | Total | - | 730 | 730 |
| | Each grid could use | - | 4 | 4 |
| Global Flux (PgC/yr) | BIO | -2.07 | -4.16 | -4.15 |
| | OCN | -2.08 | -2.33 | -2.31 |
| | FOSSIL | 9.07 | 9.07 | 9.05 |
| | AGR | 7.25 | 4.91 | 4.92 |
| Regional Flux (PgC/yr) | North America Boreal | -0.29 | -0.43 | -0.44 |
| | North America Temperate | -0.42 | -1.25 | -1.21 |
| | Tropical South America | -0.17 | -0.26 | -0.27 |
| | Temperate South America | -0.24 | -0.4 | -0.41 |
| | Northern Afirca | 0.21 | 0.32 | 0.34 |
| | Southern Africa | 0.22 | -0.3 | -0.29 |
| | Boreal Asia | -0.4 | -0.46 | -0.48 |
| | Temperate Asia | -0.3 | -0.29 | -0.27 |
| | Southeast Asia | -0.29 | -0.23 | -0.24 |
| | Australia | -0.17 | -0.4 | -0.4 |
| | Europe | -0.19 | -0.41 | -0.43 |
| independent evaluation | BIAS | 1.43 | -0.44 | -0.43 |
| | MAE | 1.92 | 1.37 | 1.35 |
| | RMSE | 2.36 | 2.11 | 2.08 |
| Deviation from the observed AGR (PgC yr$^{-1}$) | | 2.08 | -0.26 | -0.25 |

Line 209: How does GCASv2 consider the spatial representativeness errors in the inversion system?

Response: Many thanks for this question. GCASv2 do not consider the spatial representativeness errors for the GOSAT XCO$_2$ retrievals in this study. Generally, the spatial representation error must be considered when the resolution of the model grid is inconsistent with the spatial range represented by the observation data. In this study, we only use the XCO$_2$ retrievals. The reason of why we do not consider the spatial representativeness errors is that, first, the XCO$_2$ retrieval is a column averaged atmospheric CO$_2$ concentration, which is the result of full atmosphere mixing; 2) before we use the GOSAT data in GCASv2, it has been averaged within the grid cell of 1°×1°. 3) a 'super-observation' approach is adopted based on the optimal estimation theory (Miyazaki et al., 2012). A super-observation is generated by averaging all observations located within the same model grid within a DA window. Therefore, we believe that the spatial representation of the re-grided and averaged XCO$_2$ data can match the grid of the model. In addition, the model-data mismatch error of XCO$_2$ is constructed using the GOSAT retrieval error, which has been uniformly inflated by a factor of 1.9 with lowest error fixed as 1 ppm. Therefore, we did not consider the spatial representation error in this study.

Line 238: How many sites are subject to this spurious noise? Are these sites excluded from the inversion system? How much does removing data at these sites influence the inversion fluxes?

Response: We have conducted an additional assimilation for the year of 2010, in which we do not remove the spurious signals, namely all the data with the correlation coefficient with the perturbed fluxes greater than zero were used for assimilation. As shown in Table 3, on average, 87% of the observations were spurious noise and removed in this study. The spurious observations will increase the inverted global land sink and enlarge the deviation of the simulated and observed atmospheric $CO_2$ growth rate. For different TRANSCOM regions, the impact for the BIO fluxes could be in the range of -32% to 40%. We have added the following sentences in the revised manuscript (see Lines 351-355, Page 12) and added Table 3 in the revised Supporting Information.

"…Otherwise, the relationship is assumed to be spurious noise. *On average, 87% of the observations were spurious noise and removed in this study. The spurious observations will increase the inverted global land sink and enlarge the deviation of the simulated and observed AGR. For different TRANSCOM regions, the impact for the BIO fluxes could be in the range of -32% to 40% (Table S4)*. The scale of 3000 km …"

**Table 3**. Results of sensitivity tests in the year of 2010 (Wnoise is the experiment with spurious signals included)

| | | Prior | Posterior | Wnoise |
|---|---|---|---|---|
| Super Obs. Num. per window | Total | - | 730 | 730 |
| | Each grid could use | - | 4 | 28 |
| Global Flux (PgC/yr) | BIO | -2.07 | -4.16 | -4.31 |
| | OCN | -2.08 | -2.33 | -2.42 |
| | AGR | 7.25 | 4.91 | 4.67 |
| Regional Flux (PgC/yr) | North America Boreal | -0.29 | -0.43 | -0.42 |
| | North America Temperate | -0.42 | -1.25 | -1.41 |
| | Tropical South America | -0.17 | -0.26 | -0.3 |
| | Temperate South America | -0.24 | -0.4 | -0.37 |
| | Northern Afirca | 0.21 | 0.32 | 0.28 |
| | Southern Africa | 0.22 | -0.3 | -0.42 |
| | Boreal Asia | -0.4 | -0.46 | -0.33 |
| | Temperate Asia | -0.3 | -0.29 | -0.31 |
| | Southeast Asia | -0.29 | -0.23 | -0.27 |
| | Australia | -0.17 | -0.4 | -0.4 |
| | Europe | -0.19 | -0.41 | -0.28 |
| independent evaluation | BIAS | 1.43 | -0.44 | -0.41 |
| | MAE | 1.92 | 1.37 | 1.4 |
| | RMSE | 2.36 | 2.11 | 2.2 |
| Deviation from the observed AGR (PgC yr$^{-1}$) | | 2.08 | -0.26 | -0.5 |

Technical corrections:

Line 38: "BIAS" is not defined before it is used.

Response: Thanks! We have changed "BIAS" to "bias" in the revised manuscript (see Line 38, Page 2).

Line 63: "However, their carbon uptakes have significant spatial differences and interannual variations." References are needed.

Response: Thanks for this suggestion. We have added three references, namely *Bousquet et al. (2000), Takahashi et al. (2009)* and *Piao et al. (2020)*. (see Lines 65-66, Page 3)

Line 95: "However, so far, on the one hand, most studies focused on the impact of GOAST XCO 2 retrievals on the inversion of surface carbon fluxes, but in many regions, there are still large divergences for carbon sinks between different inversions with the same GOSAT data or between inversions with GOSAT and in situ observations (Chevallier et al., 2014)". Is only one study considered and cited?

Response: Many thanks for this suggestion. We have added two references in the revised manuscript, i.e., Wang et al. (2018) and Feng et al. (2016). The sentence has been revised as follows (see Line 102, page 4 in the revised manuscript):
"…between inversions with GOSAT and in situ observations (e.g., Chevallier et al., 2014; Feng et al., 2016; Wang et al., 2018), on the other hand, …"

Line 102. References are needed.

Response: Thank you! We have added two references, namely Feng et al. (2017) and Byrne et al., (2019). See Line 106, Page 4 in the revised manuscript.

Line 255: The references for the two emission inventories of FOSSIL and FIRE are out of date. ODIAC and GFEDv4 have been updated recently.

Response: We have revised the reference of ODIAC "Oda and Maksyutov (2011)" as "*Oda et al. (2018)*", and the references of GFEDv4 "van der Werf et al. (2010) and Giglio et al. (2013)" as "*Randerson et al., 2017*" (see Lines 377 and 379, Page 13)

Line 270: "The BIO carbon flux, which is the most important prior carbon flux". Why is the prior carbon flux of BIO more important than FOSSIL and FIRE to an inversion system?

Response: This statement is problematic. From the perspective of the carbon cycle, the carbon flux of terrestrial ecosystems is not more important than others. In fact, what we want to express is that because the carbon flux of terrestrial ecosystems has the greatest uncertainty and the most significant interannual variation, when using observational data to optimize surface carbon flux, the carbon flux of terrestrial ecosystems is the most concerned. We have modified that sentence to "*The BIO carbon flux, which is one of the most concerned prior carbon fluxes in an assimilation system*" in the revised manuscript. (see Line 389, Page 13)

Line 340: When the averages of the modeled and the observational values/retrievals are equal, BIAS is zero, even if all data are distant to the 1:1 line in the comparison. BIAS cannot

effectively evaluate the performance of the model by showing how much the modeled values/retrievals agree with the observational values/retrievals. The average of absolute difference between the modeled and the observational values/retrievals is needed.

Response: Thank you! We have added the mean absolute error (MAE) between the modeled and the observational values/retrievals in the revised manuscript. (see Line 532, Page 15; Lines 577-579, Page 17; Lines 599-601, Page 18; and Lines 614 – 616, Page 19)

Line 360: Does the study of Wang et al. (2019) account for the uncertainty in FOSSIL and FIRE?

Response: No, Wang et al. (2019) only optimized the terrestrial ecosystem and ocean carbon fluxes.

Line 448: What is "impact of accumulation"?

Response: As shown in the following figure (Figure 1), we find that there is a significant increasing trend for the annual BIAS between the simulated $CO_2$ concentration with the posterior flux and the observed concentration. We believe that this increasing trend is due to the accumulation of errors in the assimilation system, which may be caused by the slight overestimates of land sink in each year.

[Figure]

Figure 1. Annual mean BIAS between the surface flask observations and the simulations with posterior fluxes, the △BIAS means the difference in BIAS between two consecutive years, for example, the △BIAS in 2011 means the BIAS in 2011 minus the one of 2010.

Figures 3/4: "Biases" in the caption is easily confused with "BIAS" defined in equation 10.

Response: Thank you! We have modified the "Biases" in the caption Figures 3/4 to "BIAS". (see Line 595, page 18 and Line 605, page 19)

Table 1: BIAS cannot evaluate the performance of the model by showing how much the modeled values/retrievals agree with the observed values/retrievals.

Response: Thank you for this suggestion! According to this suggestion, we have added the mean absolute error (MAE) in Table 1 in the revised manuscript. (see Line 532, Page 15; Lines 577-579, Page 17; Lines 599-601, Page 18; and Lines 614 – 616, Page 19)

**Reference:**

Baker, D. F., et al.: TransCom 3 inversion intercomparison: Impact of transport model errors on the interannual variability of regional CO2 fluxes, 1988–2003. Global Biogeochem. Cy., 20, GB1002, doi:10.1029/2004GB002439, 2006.

Basu, S., et al.: Global CO2 fluxes estimated from GOSAT retrievals of total column CO2, Atmos. Chem. Phys., 13, 8695–8717, https://doi.org/10.5194/acp-13-8695-2013, 2013.

Bousquet, P., et al.: Regional Changes in Carbon Dioxide Fluxes of Land and Oceans Since 1980, 290 (5495), 1342-1346, https://doi.org/10.1126/science.290.5495.1342, 2000.

Byrne, B., et al.: On what scales can GOSAT flux inversions constrain anomalies in terrestrial ecosystems?, Atmos. Chem. Phys., 19, 13017–13035, https://doi.org/10.5194/acp-19-13017-2019, 2019.

Chevallier, F., et al.: Objective evaluation of surface- and satellite-driven carbon dioxide atmospheric inversions, Atmos. Chem. Phys., 19, 14233–14251, https://doi.org/10.5194/acp-19-14233-2019, 2019.

Deng, F., et al.: Combining GOSAT XCO2 observations over land and ocean to improve regional CO2 flux estimates, J. Geophys. Res. Atmos., 121, 1896–1913, https://doi.org/10.1002/2015JD024157, 2016.

Deng, F., et al.: Global monthly CO2 flux inversion with focus over North America, Tellus B, 59, 179–190, 2007.

Deng, F., et al.: Inferring regional sources and sinks of atmospheric CO2 from GOSAT XCO2 data, Atmos. Chem. Phys., 14, 3703-3727, https://doi.org/10.5194/acp-14-3703-2014, 2014.

Feng, L., et al.: Consistent regional fluxes of $CH_4$ and $CO_2$ inferred from GOSAT proxy XCH4 : XCO2 retrievals, 2010–2014, Atmos. Chem. Phys., 17, 4781–4797, https://doi.org/10.5194/acp-17-4781-2017, 2017.

Feng, L., et al.: Estimates of European uptake of $CO_2$ inferred from GOSAT $XCO_2$ retrievals: sensitivity to measurement bias inside and outside Europe, Atmos. Chem. Phys., 16, 1289–1302, https://doi.org/10.5194/acp-16-1289-2016, 2016.

Feng, L., et al.: Estimating surface $CO_2$ fluxes from space-borne $CO_2$ dry air mole fraction observations using an ensemble Kalman Filter, Atmos. Chem. Phys., 9, 2619–2633, https://doi.org/10.5194/acp-9-2619-2009, 2009.

Gurney, K. R., et al.: Towards robust regional esti-mates of CO2 sources and sinks using atmospheric transport models, Nature, 415, 626–630, https://doi.org/10.1038/415626a, 2002.

Gurney, K. R., et al.: TransCom 3 CO2 inversion intercomparison: 1. Annual mean control results and sensitivity to transport and prior flux information, Tellus B, 555–579, 2003.

Hu, L., et al.: Enhanced North American carbon uptake associated with El Niño, Sci. Adv., 5, eaaw0076, https://doi.org/10.1126/sciadv.aaw0076, 2019.

Kang, J.-S., et al.: Estimation of surface carbon fluxes with an advanced data assimilation methodology, J. Geophys. Res., 117, D24101, https://doi.org/10.1029/2012JD018259, 2012.

Liu, J., et al.: Contrasting carbon cycle responses of the tropical continents to the 2015–2016 El Niño, Science, 358, eaam5690, https://doi.org/10.1126/science.aam5690, 2017.

Liu, Y., et al.: Estimating surface carbon fluxes based on a local ensemble transform Kalman filter with

a short assimilation window and a long observation window: an observing system simulation experiment test in GEOS-Chem 10.1, Geosci. Model Dev., 12, 2899–2914, https://doi.org/10.5194/gmd-12-2899-2019, 2019.

Miyazaki, K., et al.: Assessing the impact of satellite, aircraft, and surface observations on CO2 flux estimation using an ensemble-based 4-D data assimilation system, J. Geophys. Res., 116, D16306, https://doi.org/10.1029/2010JD015 366, 2011.

Monteil, G., et al.: The regional European atmospheric transport inversion comparison, EUROCOM: first results on European-wide terrestrial carbon fluxes for the period 2006–2015, Atmos. Chem. Phys., 20, 12063–12091, https://doi.org/10.5194/acp-20-12063-2020, 2020.

Nassar, R., et al.: Inverse modeling of $CO_2$ sources and sinks using satellite observations of CO2 from TES and surface flask measurements, Atmos. Chem. Phys., 11, 6029–6047, https://doi.org/10.5194/acp-11-6029-2011, 2011.

Niwa, Y., et al.: Imposing strong constraints on tropical terrestrial CO2 fluxes using passenger aircraft based measurements, J. Geophys. Res., 117, D11303, doi:10.1029/2012JD017474, 2012.

Oda, T., et al.: The Open-source Data Inventory for Anthropogenic CO2, version 2016 (ODIAC2016): a global monthly fossil fuel CO2 gridded emissions data product for tracer transport simulations and surface flux inversions, Earth Syst. Sci. Data, 10, 87–107, https://doi.org/10.5194/essd-10-87-2018, 2018.

Peters, W., et al.: An atmospheric perspective on North American carbon dioxide exchange: CarbonTracker, P. Natl. Acad. Sci., 104, 18925–18930, https://doi.org/10.1073/pnas.0708986104, 2007.

Peters, W., et al.: Seven years of recent European net terrestrial carbon dioxide exchange constrained by atmospheric observations, Glob. Change Biol., 16, 1317–1337, https://doi.org/10.1111/j.1365-2486.2009.02078.x, 2010.

Piao, S., Wang, X., Wang, K., et al.: Interannual variation of terrestrial carbon cycle: Issues and perspectives, Glob Change Biol., 26, 300– 318, https://doi.org/10.1111/gcb.14884, 2020.

Randerson, J.T., et al.: Global Fire Emissions Database, Version 4.1 (GFEDv4). ORNL DAAC, Oak Ridge, Tennessee, USA. https://doi.org/10.3334/ORNLDAAC/1293, 2017.

Rödenbeck, C., et al.: Time-dependent atmospheric CO2 inversions based on interannually varying tracer transport. Tellus B, 55: 488-497. https://doi.org/10.1034/j.1600-0889.2003.00033.x, 2003.

Rödenbeck, C.: Estimating CO2 sources and sinks from atmospheric mixing ratio measurements using a global inversion of atmospheric transport, Technical Report 6, Max Planck Institute for Biogeochemistry, Jena, 2005.

Scholze, M., et al.: Mean European carbon sink over 2010–2015 estimated by simultaneous assimilation of atmospheric CO2, soil moisture, and vegetation optical depth. Geophysical Research Letters, 46, 13796– 13803. https://doi.org/10.1029/2019GL085725, 2019.

Takahashi, T., et al.: Climatological mean and decadal change in surface ocean pCO2, and net sea-air CO2 flux over the global oceans. Deep Sea Research Part II: Topical Studies in Oceanography, 56 (8-10): 554-577, https://doi.org/10.1016/j.dsr2.2008.12.009, 2009.

Wang, J. S., et al.: A global synthesis inversion analysis of recent variability in CO2 fluxes using GOSAT and in situ observations, Atmos. Chem. Phys., 18, 11097–11124, https://doi.org/10.5194/acp-18-11097-2018, 2018.

Zhang, H. F. et al. Net terrestrial $CO_2$ exchange over China during 2001–2010 estimated with an ensemble data assimilation system for atmospheric CO2. J Geophys Res 119, 3500–3515, 2014.